# A New Methodological Approach Integrating Motion Capture and Pressure-Sensitive Gait Data to Assess Functional Mobility in Parkinson’s Disease: A Two-Phase Study

**DOI:** 10.3390/s25195999

**Published:** 2025-09-29

**Authors:** Sabrina Köchli, Isabel Casso, Yvonne N. Delevoye-Turrell, Stefan Schmid, Dawn C. Rose, Caroline Whyatt

**Affiliations:** 1School of Music, Lucerne University of Applied Sciences and Arts, 6010 Lucerne, Switzerland; dawn.rose@hslu.ch; 2UMR 9193–SCALab–Sciences Cognitives et Sciences Affectives, Université de Lille, CNRS, F-59000 Lille, France; isabel.casso@univ-lille.fr (I.C.); yvonne.delevoye@univ-lille.fr (Y.N.D.-T.); 3School of Health Professions, Bern University of Applied Sciences, 3007 Bern, Switzerland; stefan.schmid@bfh.ch; 4Faculty of Medicine, University of Basel, 4003 Basel, Switzerland; 5Sport and Geography, Department of Psychology, University of Hertfordshire, Hertfordshire AL10 9AB, UK; c.whyatt@herts.ac.uk

**Keywords:** Parkinson’s disease, functional mobility, motion capture, gait mat analysis, rehabilitation

## Abstract

**Highlights:**

**What are the main findings?**
We report the development of a novel measure of functional mobility (FMA-P) that provides detailed insight into symptom-related movement impairments in Parkinson’s disease.By integrating motion capture and pressure-sensitive gait mat technology, the FMA-P evaluates multiple aspects of mobility—including balance, posture, gait, sit-to-stand transitions, turning, and reaching—that are often overlooked by standard assessments focused primarily on task duration.

**What is the implication of the main findings?**
Specific movement tasks, particularly those involving yaw rotation, are sensitive indicators of changes in Parkinson’s disease symptom severity.Rehabilitation programs should prioritize these tasks, as targeting them may optimize functional gains and better monitor disease progression.

**Abstract:**

Existing clinical assessments of Parkinson’s disease (PD) primarily focus on stratifying symptom severity or progression rate, which limits their ability to capture changes in functional mobility—an important factor in evaluating rehabilitation outcomes. To address this gap, we developed a novel methodology, the Functional Mobility Assessment for Parkinson’s (FMA-P), which integrates motion capture and pressure-sensitive gait analysis to explore key aspects of functional mobility. Study 1. To develop the FMA-P, we conducted a pilot study involving 12 individuals with PD and 12 age-matched healthy controls, who each completed the FMA-P sequence three times. The sequence included the following tasks: rising from a chair, walking through a doorway, turning, bending to pick up and place an object, and returning to a seated position. Results from Study 1 demonstrated that the FMA-P is a sensitive tool for identifying functional impairments in PD. In particular, significant differences between people with Parkinson’s (PwP) and controls were observed during chair rise (higher peak trunk inclination, *p* = 0.006; lower mean trunk jerk, *p* = 0.003) and turning task (longer task duration, *p* = 0.026 and lower mean heel strike angle, *p* = 0.007), providing critical insights into postural stability. Study 2. To assess changes in functional mobility over time, we conducted a 12-week repeated-measures intervention study with 12 participants with PD. Results from Study 2 indicated notable improvements in turning stability and balance. Participants demonstrated reduced turning time (*p* = 0.006) and increased yaw rotation in the head (*p* = 0.001), trunk (*p* = 0.002), and pelvis (*p* = 0.012). In contrast, no significant changes were observed in standard clinical measures (i.e., Timed Up and Go and task duration). The FMA-P offers fine-grained insights into movement quality, making it a valuable tool for early diagnosis, monitoring intervention efficacy, and guiding rehabilitation strategies in individuals with PD.

## 1. Introduction

Converging evidence from global surveys suggests that the prevalence of people with Parkinson’s (PwP) will double by 2050 [1,2]. While PD is well known to occur in older adulthood, there is an increase in young-onset PD, with a prevalence of 3–5% of people affected before the age of 40 years [3,4]. However, the differential diagnosis for PD in younger people may not be considered by physicians [5,6]. Inadequate differential diagnoses and measurement protocols that could detect PD in its early stages result in delays in treatment that negatively impact the quality of life for PwP and their care partners [6,7]. In recent years, advanced neuroimaging approaches, including diffusion tensor imaging, neuromelanin-sensitive magnetic resonance imaging (MRI), and iron-sensitive sequences, have been investigated as potential biomarkers to enable earlier detection and differentiation from atypical parkinsonian syndromes [8,9,10].

Parkinson’s disease (PD) is a neurodegenerative brain disorder caused by a loss of dopaminergic cells in the basal ganglia, resulting in involuntary or uncontrollable motor movements [1,2,3]. Motor symptoms are characterized by tremor, rigidity, slowness of movement (i.e., bradykinesia) and difficulties with gait, and are often accompanied by non-motoric symptoms such as impaired cognition and psychological behaviors [4,11,12]. These clinical manifestations are directly relevant to rehabilitation, as they strongly affect mobility, balance, and independence in daily life, and have been shown to respond to structured rehabilitation interventions in PD [13,14,15].

There are several assessment scales for the evaluation of PD [16]. The gold standard scale for assessing the severity and progression of PD is the Movement Disorder Society-sponsored revision of the Unified Parkinson’s Disease Rating Scale (MDS-UPDRS), which relies on trained clinicians to evaluate the presence and severity of motoric and non-motoric symptoms [17]. The MDS-UPDRS, an expanded version of the Unified Parkinson’s Disease Rating Scale (UPDRS), was developed to capture a range of clinically relevant issues in PD, including problems related to daily living and non-motor symptoms that were insufficiently captured in the initial version [17]. The Hoehn & Yahr scale, integrated into the MDS-UPDRS, is commonly used to identify progressive stages of PD, with Stage 1 indicating no functional disability, and Stage 5 confinement to a wheelchair [18].

Clinical rating scales and self-report questionnaires are also important instruments for assessing the severity, stage, and progression of PD, and the impact of motor symptoms on activities of daily living [19]. Examples include the Parkinson’s disease Activities of Daily Living Scale (PADLS) [20], and the Parkinson’s Disease Questionnaire (PDQ−39; [21]), which includes specific subscales for mobility and activities of daily living. The MDS-UPDRS Part II and PADLS correlate with examination-based measures of motor symptom severity [22]. Despite their utility, such clinical rating scales and assessment tools for PD face significant challenges [23]. The complexity of PD, and the potential for participant demand bias, can lead to rating scales underestimating the prevalence and severity of motor and non-motor symptoms, limiting their accuracy and reliability. This challenge is further compounded by the fact that motor disabilities may not be clinically apparent at the early stages of the disease [24,25].

Action-based observation tools that focus on physical performance are often recommended for the assessment of functional difficulties with gait, postural balance, increased risk of falls and reduced mobility for PwP [11,26]. For example, the Timed Up and Go (TUG) test [27] is a commonly used clinical tool that has been used as for predicting risk of falls in PwP [28]. The sequence involves standing up from a seated position, walking a prescribed distance, turning, and returning to a seated position. However, the sensitivity of the TUG, particularly in early-stage Parkinson’s disease, has been questioned [29]. A significant limitation of such tools is their emphasis on quantifying the ‘end product’ of an action, such as the time taken to complete a task. While Bradykinesia is a cardinal feature of PD, this summative product-oriented approach fails to capture the nuances of how a movement unfolds and thus, fails to provide more granular insight into which motor difficulties are apparent and, consequently, which rehabilitation strategies would be appropriate.

To provide more fine-grained insight into the quality of motion, several recent studies have investigated gait performance using advanced technology such as inertial movement sensors, optical motion capture, or pressure-sensitive gait mats. These tools allow the assessment of clinically relevant parameters, such as velocity, step length, cadence, and gait asymmetries in PwP [16,30,31,32,33]. Deconstructing performance on the TUG through the amalgamation of wearable technology has demonstrated variation in cadence, angular velocity of arm-swing, turning duration, and time to perform turn-to-sits as important factors in early detection of PD [29,34]. Further, this level of precision enables robust profiling of bradykinesia associated with movement amplitude and rhythm, which may be sensitive to medication use in PD [35]. Such robust instrumental profiling has also provided insight into the nuanced differences in turning when walking in PwP [29], with evidence suggesting that turning, in particular, may be especially sensitive for detecting early changes in PD [36,37].

Inertial sensors and wearable technologies offer clear advantages in kinematic measurement, including independence from lighting conditions, minimal occlusion issues, and portability for use in diverse environments [38,39]. Nevertheless, these systems rely on extrapolating positional information from acceleration signals, a process that is complex and prone to cumulative error due to sensor noise and drift [40,41,42,43]. As a result, they provide limited insight into fine-grained, whole-body kinematics, which is essential for the development of robust profiling protocols. In contrast, fixed optical motion capture remains the gold standard for quantifying nuanced movement dynamics in controlled environments, enabling the detection of subtle alterations in performance even in early-stage PD [38,44,45]. Establishing such features within a robust reference framework creates the foundation for subsequent translation to lower-cost wearable technologies, thereby providing a pathway for wider clinical application. Studies vary widely in design and outcome metrics, making it unclear which spatiotemporal gait parameters are most clinically relevant for PwP [46,47].Although wearable technology is expected to further enhance profiling precision and expand our understanding of PD kinematics [48,49], there is currently little consensus regarding optimal measurement protocols.

In terms of therapeutic management, deep brain stimulation (DBS) is an established intervention for advanced stages of PD, offering significant improvements in tremor, rigidity, and motor fluctuations [50,51]. However, DBS is invasive, costly, and primarily considered when pharmacological treatment no longer provides sufficient symptom control [52,53]. It may also be associated with surgical risks and neuropsychological side effects, but careful patient selection, including MRI-based diagnostic approaches, helps optimize outcomes by enabling precise identification of target structures for DBS [54,55,56,57,58]. In contrast, non-invasive functional mobility training offers a safe alternative across disease stages, directly targeting gait, postural control, and daily activities, potentially delaying the need for invasive procedures [59].

Music- and movement-based interventions are a particularly promising form of functional mobility training [60]. Approaches such as rhythmic auditory stimulation and dance directly improve gait, balance, and movement synchronization [61,62]. Dance-based programs, such as tango or adapted ballroom dance, provide additional benefits for balance, mobility, and quality of life [63,64]. A meta-analysis further confirmed that music-based movement therapy can enhance walking ability, balance, and overall quality of life in PwP. Beyond motor improvements, these interventions enhance adherence and psychosocial well-being, complementing conventional physiotherapy and addressing both motor and non-motor symptoms of PD [65].

Our aim was to address a current gap in the literature, namely the lack of standardized, multidimensional tools that capture fine-grained kinematic aspects of functional mobility in PwP. Existing assessment methods, including clinical rating scales and wearable sensors, often fail to detect subtle movement impairments, particularly in early-stage PD. To fill this gap, we developed a way of using kinematic measures through a sequence of movements that could differentiate between PwP and controls, and to enable the monitoring of (potential) changes over time, for example, as a result of an intervention designed to improve functional mobility for PwP.

Against this background, we developed a novel protocol, the Functional Mobility Assessment for Parkinson’s (FMA-P), based on a sequence of movements associated with daily activities and commonly used in clinical measures (such as the TUG). The FMA-P was conceptually derived from the Physiological Laboratory Mobility (PLM) framework, which guided our selection of movement tasks and metrics [66]. Moreover, the FMA-P employs a multidimensional approach by integrating data from a motion capture system and a pressure-sensitive gait mat with additional qualitative observational metrics (i.e., an associated Performance Score). This integrated approach offers a novel way to capture subtle kinematic changes and functional deficits, enabling sensitive differentiation between PwP and healthy controls and providing a tool for evaluating the effectiveness of interventions.

Here we present two studies: the first is a cross-sectional mixed methods study describing in detail the development process of the FMA-P that helped establish which metrics most reliably capture problems with functional mobility in PD, and the second is a repeated measures study demonstrating the application of the FMA-P in the concept of a music- and movement-based intervention study designed to improve functional mobility for PwP.

## 2. Materials and Methods

This research was conducted by an international team for which data was collected in both Switzerland and the UK. These studies form part of a wider study designed to co-develop a new music-and-movement-based intervention–Songlines for Parkinson’s—designed to improve functional mobility for PwP. The draft intervention protocol was preregistered on the Open Science Framework (OSF), and its developmental process has been reported previously [67].

The overall goal of this research was to develop and evaluate a multidimensional assessment tool—the FMA-P—that would be suitable both for distinguishing PwP from healthy controls and for detecting changes in functional mobility over time, for example, following a structured intervention. We used a two-phase study design:Study 1 was a cross-sectional, mixed-methods study primarily focused on the development of the FMA-P, alongside its preliminary evaluation.Study 2 applied the FMA-P in a repeated-measures design to evaluate its sensitivity to change following an intervention program.

### 2.1. Participants and Ethical Considerations

In Switzerland, participants were recruited via a Parkinson’s information day held at the local hospital (Luzerner Kantonsspital) and through information shared about the study by Parkinson Schweiz. In the UK, participants were recruited through established researcher networks and from talks given about the project at Parkinson’s UK groups in the local area (Hertfordshire). Participants were eligible if they had been formally diagnosed with PD, were independently mobile (up to stage 4 on the Hoehn & Yahr scale [18]), were between the ages of 40 and 80, and had a stabilized medication regimen. Controls were also aged between 40 and 80, with no other motor or neurological impairments. Additionally, we administered the mini MoCA [68] to the participants with PD to check for potential cognitive impairments (i.e., did not score below 12).

In Study 1, a total of 12 PwP and 12 healthy controls participated (the sample of PwP was split equally between the UK and Switzerland, whereas for the control group, nine participated in Switzerland and 3 in the UK). For Study 2, a total of 12 PwP participated in a music-based intervention study in the UK. Measurements were taken at three timepoints: baseline (4–6 weeks prior to the intervention), pre-intervention (1–2 weeks prior) and post-intervention (1–2 weeks after completion).

Both studies were conducted according to the guidelines of the Declaration of Helsinki and the Guidelines of Good Clinical Practice [69]. The Ethics Committee of Lucerne University of Applied Sciences and Arts approved the parts of this study in Switzerland (Protocol Number EK-HSLU 002 M 22, date of approval 16 March 2022). The Ethics Committee of the Health, Science, Engineering & Technology (ECDA) of the University of Hertfordshire approved the parts of this study that took place in the UK (Protocol Reference LMS/PGR/UH/04935, date of approval 31 March 2022). All participants received detailed information on the study procedures, which were approved by the respective ethics committees. Written informed consent was obtained before any measurements.

### 2.2. Materials

#### 2.2.1. Motion Capture System

Kinematic movement sequences were assessed using Vicon Motion Capture Systems (Vicon Motion Systems Inc, Los Angeles, CA, USA), consisting of eight cameras in Switzerland and 12 cameras in the United Kingdom. Due to room size constraints in the United Kingdom, four more cameras were used in the corners of the room to capture all the markers during the standing up and the turning task. The Vicon systems were controlled by the Nexus software (version 2.15, Vicon United Kingdom, Oxford, UK) and recorded with a sampling frequency of 100 Hz. A full-body model (Plug-in-Gait, Vicon Motion Systems) was used to record and quantify movement parameters relevant to PD, complemented by the Conventional Gait Model 2 (CGM2), which included additional markers on the anterior thigh and shank to enhance measurement of lower-limb kinematics. Marker placement followed the standard protocols and is illustrated in Appendix A [70,71].

#### 2.2.2. Walkway Gait Analysis System

A five-meter pressure-sensitive Zeno walkway was used to record locomotion and foot placement parameters (ProtoKinetics LLC, Havertown, Pennsylvania). The surface of the Zeno gait mat was marked with tape to define the walking distance and turning points (with an active area of 4.88 m × 1.22 m). Data was recorded at a sampling frequency of 100 Hz and later analyzed with PKMAS (ProtoKinetic Movement Analysis Software, version 6.00c3). The PKMAS software was synchronized with the Vicon Nexus software so that the recording times were simultaneous (controlled by the PKMAS software).

### 2.3. Study 1: Development and Pilot Testing of the Functional Mobility Assessment for Parkinson’s (FMA-P)

The overarching research question of Study 1 was, first, to determine which biomechanical parameters captured by the FMA-P most effectively distinguish PwP from healthy controls, particularly in movement tasks that involve transitions between habitual and goal-directed actions; and second, to assess how the sensitivity of the FMA-P compares to that of traditional clinical tools such as the TUG test.

Based on this research question, the following hypotheses were formulated:

**H1:** 
*The FMA-P protocol captures distinct kinematic and spatiotemporal parameters that significantly differ between PwP and healthy controls.*


**H2:** 
*The FMA-P demonstrates greater sensitivity than the TUG in detecting subtle impairments in motor control among PwP.*


**H3:** 
*The most pronounced group differences will be observed in FMA-P tasks that require shifts between habitual and goal-directed motor behavior.*


#### 2.3.1. Development of the FMA-P Study Protocol

The FMA-P was conceptually developed based on the Postural-Locomotion-Manual Test (PLM), which aimed to qualitatively and quantitatively evaluate various factors that characterize movement abilities in PwP [66]. The PLM provided a three-step movement sequence that mimicked typical activities of daily living, i.e., lifting, transporting, and placing an object. It comprised three distinct phases: the postural phase (*P*), the locomotion phase (*L*), and the manual phase (*M*). To quantify performance, the authors suggested using a Simultaneity Index (*SI*), which was calculated as follows:(1)$$SI=\frac{P+L+M}{MT}$$
where *MT* represents the total movement time for the task, measured from the initial grasping of the object to its placement, *P* was defined as the time from grasping the object until the body was upright, *L* was the time of locomotion/walking, and *M* was the time for forward movement of the arm to aim and place the object. A lower *SI* thus reflected increased movement time and greater functional motor disability [72].

The PLM, developed in the 1980 s, could be considered ahead of its day. Although a 2013-paper [73] reported the protocol to correlate fairly with the UPDRS, it was not picked up as systematically by Parkinson’s researchers (probably due to the arduous nature of processing that type of data at that time), and is no longer in use (personal communication with the original authors, 25.05.2020). Nevertheless, the architecture of the PLM can be utilized as a basis for the proposed protocol, with objective kinematic parameters simultaneously extracted from motion capture and gait mat recordings. To further develop the PLM and enhance its applicability for PwP, three additional mobility measures were included: transitioning between sitting to standing (and vice versa) and a 6 × 3 m walk with turn, tasks that form part of the TUG. These tasks allow for comparison with TUG performance and with other studies that report commonly used kinematic measures.

A key challenge with the TUG is the reliance on time-based metrics that primarily reflect automatic, habitual movements such as walking and transitioning between states. Previous neurological evidence suggests that dopaminergic neuron loss in PD disrupts the balance between habitual and goal-directed actions [74,75]. Specifically, degeneration of the putamen impairs habitual movement, increasing reliance on goal-directed strategies mediated by the caudate nucleus and prefrontal cortex [75]. This compensatory reliance on goal-directed control is thought to contribute to cardinal features such as bradykinesia. However, experimental findings are inconclusive, with evidence both for a decline in habitual movement [76] and increased reliance on habitual control [77], possibly due to methodological differences (e.g., task complexity, implicit vs. explicit learning protocols). The impact of early-stage PD on habitual actions remains uncertain and may be mediated by disease severity-dependent deficits in goal-directed behavior [78]. This interplay may also underlie subtle changes in goal-directed gait [79] and turning in PwP [80].

Given this ambiguity, and to better assess both automatic and goal-directed tasks with ecological validity, we collaborated with PwP to include a functional goal-directed task (picking up and placing a set of keys onto a hook after a turning sequence) into the FMA-P, which requires high precision and planning. Integrating both protocols may allow for a more comprehensive profiling of motor control while providing a framework for future research. Finally, a ‘doorway’ was incorporated to assess potential freezing episodes during locomotion, a common phenomenon in PwP [81,82]. Figure 1 shows a diagram depicting the newly developed FMA-P measurement protocol. For preparation of the measurement, a height-adjustable, armless chair was placed on the start of the gait mat. Care was taken to ensure that the participants maintained a 90-degree angle at the knee. A doorframe was placed in the middle of a three-meter walkway (at 1.5 m from the start). The key was placed 0.75 m in front of the door frame, directly after the turning point (three meters from the start). Participants were free to turn in either clockwise or counterclockwise direction; the placement of the key was aligned with their dominant hand to facilitate a natural, comfortable turning movement. To account for the directional impact on kinematic features, all turning-related measures were normalized or sign-adjusted during data processing, ensuring comparability across directions and minimizing any influence on subsequent analyses.

As shown in Figure 1, the FMA-P protocol was developed to include specific tasks adapted from clinical measures and integrated into one sequence. These tasks are described in detail in the following section.

Sit-to-stand and Stand-to-sit

Sit-to-stand and stand-to-sit are crucial components of daily independent living and consequently a key variable influencing the quality of life [83]. Studies have indicated that 81% of PwP experience difficulty rising from a chair [84], and although most studies do not specifically address sitting down, it is likely that PwP face similar challenges during this process. These transitions are commonly assessed with time-based clinical scales such as the Berg Balance Scale (BBS; [85]) and the Timed Up and Go (TUG; [86]), where PwP typically show prolonged transition times [87,88].

Biomechanical studies suggest altered motor strategies in PwP, such as exaggerated hip flexion and reduced knee extension, reflecting impaired anticipatory control of the center of mass [87,89].

Turning

Due to impaired postural control, PwP tend to take small steps and make frequent foot adjustments to maintain balance when changing direction [90]. Studies have demonstrated that PwP exhibit simultaneous rotation of the head, thorax, and pelvis during turning, whereas healthy individuals follow a cranial to caudal sequence [91,92]. This disrupted axial coordination contributes to turning difficulties and an increased risk of falls risk [93,94].

Functional reach

The basal ganglia, which are most affected in PD, play a crucial role in coordinating postural control and voluntary movement [95]. Functional mobility actions such as reaching, grasping, and placing objects are frequently impaired in PwP and have been linked to an increased risk of falls [90,96]. These activities rely on different spinal pathways to integrate complex movement with postural stability, making them a sensitive marker of movement impairments in PD [72,97].

Although the UPDRS Part III is widely used to assess motor function in PwP, its ratings can be inconsistent across evaluators and often lack ecological validity for goal-directed postural control, raising questions about which specific aspect(s) of postural control are captured [98]. The Functional Reach Test or FRT (not to be confused with the current functional reach task), has been shown to correlate more closely with dynamic balance during goal-directed actions, such as reaching for objects at different heights [99,100]. To ensure ecological validity, the FMA-P therefore includes a goal-directed reaching task developed in close collaboration with PwP. After piloting different objects, participants identified picking up a key from the floor and placing it on a hook at shoulder height as a natural yet challenging everyday activity.

Before initiating the task, handedness (left or right) was documented, and placement was adjusted to align with the dominant hand and shoulder height, ensuring consistency in execution and data analysis.

#### 2.3.2. Additional Tasks

As part of the FMA-P protocol, a set of additional tasks addressing key features of PD, such as posture, gait, and freezing, was included and integrated into the biomechanical analysis but was administered separately from the FMA-P sequence. The specific tasks are described in detail in the following section:Standing upright

Ten seconds of upright standing were included as an additional task to assess general postural stability. This task was included to quantify postural tilt and assess the center of pressure (COP) displacement.

Locomotion

Gait is a complex biomechanical task vital to independent functioning in everyday life. A large portion of walking (75%) tends to occur in bouts of <40 steps punctuated by brief rests [101].

Parkinsonian gait is characterized by short steps, narrow-based flexed knees, and stooped posture, which can serve as markers of disease progression and risk of falls [90]. Key biomechanical markers include stride length/variability, stride velocity, toe-off (TO), and heel-strike (HS) angles [32,102,103]. Previous studies indicate that healthy controls exhibit higher HS angles and TO angles, compared to PwP, highlighting the biomechanical gait differences [104,105].

While gait analysis often focuses on lower-body dynamics, it involves upper-body coordination and dynamic equilibrium. Reduced arm swing and altered acceleration profiles are sensitive markers for distinguishing between PwP and healthy controls [46,106] and differentiate between PwP who are likely to be classified as ‘fallers’ vs. ‘non-fallers’ [107,108,109]. In the FMA-P, continuous gait was assessed on a five-meter gait mat, including turning, with participants asked to walk as fast as possible to capture these parameters.

Freezing

In PD, freezing of gait (FOG) is one of the most disabling locomotor symptoms, characterized by an involuntary “sticking” of the feet to the floor, which significantly increases the risk of falls [110,111]. A recent study reported a weighted prevalence of FOG at 50.6% in 9071 PwP [112]. Research has demonstrated that FOG can be triggered by narrow spaces or doorways [81]. To address this in our adaptation of the PLM, we incorporated a doorway (dimensions 1.22 m × 2.02 m) into the locomotion component to capture data in relation to “freeze-like” events (FLE).

#### 2.3.3. FMA-P Performance Score

Whilst developing the FMA-P, it became apparent that participants used various strategies to perform the tasks. For example, participants might offset their feet to assist in standing, or swing one or both arms, or use one or both hands to assist in standing from a seated position. Recording and analyzing these behavioral strategies provides important information on how each task is completed and allows us to translate motion capture and gait mat data into clinically meaningful metrics that can guide rehabilitation.

Consequently, to ensure the tasks performed in the FMA-P sequence were captured at a performative level, we additionally developed a systematic observational record to be administered alongside the FMA-P performance (Appendix A).

The Performance Score is a list of seven functional mobility requirements that can be recorded while administering the FMA-P (10 items in total, plus four descriptions). A four-point ordinal scale (0 indicates no problems with the task, 3 indicates difficulties with the task) is used to assess the quality of performance with a maximum total score of 30. The higher the score, the more functional mobility is impaired.

#### 2.3.4. Timed up and Go (TUG)

The TUG protocol is a clinical assessment of mobility and fall risk in which an individual rises from a seated position, walks 3 m, turns, returns, and sits back down, with time-to-completion used to evaluate functional mobility, balance, and gait stability [27]. The TUG protocol was included to provide comparison data for the functional mobility aspects of the newly developed FMA-P (i.e., sit-to-stand, turning, return to seated). The TUG was administered prior to the FMA-P but after the additional tasks described in Section 2.3.2.

#### 2.3.5. The WHO−5 Measure of Wellbeing

The WHO−5 is a generic rating scale of subjective wellbeing [113,114,115]. The five statements refer to the past two weeks, are positively worded, and are scored using a 6-point Likert scale. An acceptable Cronbach’s alpha coefficient has been reported for this measure (>0.80, [116]).

#### 2.3.6. Study Design and Procedure

A 2 × 2 mixed factorial design was employed, with Group (PwP vs. Control) as a between-subjects factor and Protocol (TUG vs. FMA-P) as a within-subjects factor. Biomechanical performance was assessed across multiple tasks (e.g., sit-to-stand), with predefined biomechanical parameters extracted for each task. Each parameter was analyzed independently to assess group differences.

Upon arrival, participants were asked to fill out the WHO−5 questionnaire; then the experimenter proceeded to administer the MDS-UPDRS part III (Motor Examination), including the Hoehn & Yahr clinical measure [18].

Demographic and anthropometric data (i.e., age, sex, body weight, and body height) were provided by participants. Foot length, shoulder offset, elbow width, wrist width, and hand thickness were measured on-site using a caliper. Anthropometric data were used to create a model of the participant within Vicon Nexus software.

The gait mat and motion capture data were recorded during the following tasks performed by PwP. The experimental procedure is illustrated in Figure 2.

Participants were first asked to stand upright for at least 10 s, then to walk as fast as possible six consecutive times on the gait mat, covering a total distance of approximately 18 m (not including the turn on the mat).

After the initial demonstration of the TUG by the experimenters, participants were instructed to perform the TUG themselves over three consecutive times [27]. Once finished, the experimenter conducted the demonstration for the FMA-P, and participants were asked to perform the FMA-P three consecutive times. The instructions for both TUG and FMA-P were to walk and complete each task as fast as possible; both protocols were performed within the same setting (e.g., same chair, same distance to the turning task, and the same technologies were applied). Short breaks were provided between tasks to minimize fatigue, and the tasks were ordered according to complexity, so that the FMA-P followed the TUG in a way that made it easier for participants to understand and perform.

#### 2.3.7. Data Processing and Functional Mobility Analysis

All biomechanical metrics described in the present section were calculated using 3D motion capture data. The analysis of the COP was performed using data collected with the gait mat. The algorithms for metric calculation are available in a public repository linked in the Appendix A.

Recorded trials were reconstructed in three-dimensional coordinates for each marker. Using the Vicon Nexus software, markers with missing data were observed. The trajectory editor tool automatically identified gaps in marker trajectories. Polynomial gap filling was applied for gaps ≤ 10 frames and applied Vicon Nexus gap filling algorithms for >10 frames (e.g., rigid body fill, spline fill, and pattern fill). An additional data cleaning process consisted of mitigating spikes of aberrant data by applying a moving average technique of up to 15 samples (0.15 s). Data were excluded in cases of marker occlusion and/or inappropriate interpolation.

Data from the Zeno walkway system was pre-processed using the default parameters in the PKMAS software. In the case of COP, raw data was filtered using a 5-pole, low-pass Butterworth Filter with zero lag at 10 Hz, as advised in the PKMAS manual.

Missing data were handled using a multiple imputation procedure clustered either by experimental group (PwP, Control) and protocol pairs (TUG, FMA-P) or by intervention phase (Baseline, Pre, Post). Within each cluster, five imputed datasets were generated using the predictive mean matching method (PMM). A single, complete dataset was then created for analysis by pooling these imputations, which involved averaging the five imputed values for each missing data point. Multiple imputation is recommended when the percentage of missing data is between 5% and 20%. For missing data above 20%, it is necessary to consider whether multiple imputation could bias the results [117]. The mean percentage of missing data across all biomechanical parameters was 11% for Study 1 and 4% for Study 2.

All biomechanical and gait metrics analyzed in this study are summarized in Table 1, including calculation methods, associated tasks, expected differences between PwP and controls, and predictive relevance. The following sections describe task segmentation and key procedural notes.

Sit-to-stand and Stand-to-Sit

The sit-to-stand task was defined as the phase for which the upper body bent and ended at the initiation of the first footstep, while the stand-to-sit began when the turning foot made contact, initiating an upper-body bend.

Peak pelvis flexion, calculated as the minimum angle between the trunk-pelvis and trunk-knees vectors [120].

Knee flexion, defined as the minimum angle between the knee-pelvis and knee-heel vectors, was also assessed. Greater angles indicate reduced flexion; we hypothesized this angle would be greater in PwP, signaling a need for greater postural stability, while healthy controls would exhibit lower flexion angles [87].

Turning

During the turning movement, the coordination, and thus the turning strategy, across three segments of the body: head, trunk, and pelvis was assessed.

The turning task duration was calculated from the last contact of the turning foot to the first contact of the swing foot after completing a 180-degree turn, with PwP expected to take longer. Turning movement smoothness was assessed via mean jerk of the pelvis, hypothesizing reduced jerk in PwP, indicating less dynamic movement to compensate for poor postural control.

Total angular displacement during the turn using absolute yaw rotations (i.e., rotation around the vertical axis or side-to-side) for each segment was quantified. The maximum trunk inclination in anteroposterior (AP) and mediolateral (ML) directions was calculated, hypothesizing greater inclination in PwP, especially in the ML axes. Mean COP displacement in AP and ML directions was used to assess stability.

Functional reach

The key pick task duration was defined as the time from the first upper body movement (initiation of bending) to when the key was lifted, with PwP expected to take longer to complete the task.

Previous research by Stack et al. (2006) indicated that severe cases of PD may rely on a diagonal reaching strategy; however, these findings are somewhat confounded, as the study was conducted in a naturalistic environment (e.g., a kitchen counter), where some participants made use of fixed support during the task [96].

Bending was quantified by maximum pelvis and knee flexion angles. We hypothesized that PwP would exhibit reduced pelvis flexion and greater knee flexion than controls, suggesting compensatory strategies to maintain balance [121].

Stance stability was assessed by AP foot distance (between toe markers in the sagittal plane), feet aperture angle, and the maximum stance width. The aperture angle was defined as the angle formed by the vectors connecting the toes to the heels of each foot, and the maximum stance width was defined as the largest horizontal distance between the two-foot segments. PwP were expected to show wider stance angles and greater stance width to improve their stability [122].

The maximum toe off angle was calculated, with PwP predicted to have smaller angles, indicating controlled foot movement. Mean COP displacement in AP and ML directions was used to profile general stability, with PwP expected to show reduced displacement.

Standing upright

Postural sway and jerk are key indicators of balance control. Sway displacement is greater in elderly fallers [123], while PwP exhibit less smooth postural acceleration and jerk [88,124]. Postural sway using COP displacement in the AP and ML directions was assessed via a gait mat. Root mean square (RMS) values for pelvis and trunk acceleration were also calculated. We hypothesized that PwP would show greater trunk inclination and COP displacement in both directions due to impaired stability alongside increased trunk acceleration and greater trunk and pelvis jerk. Upper body lateral tilt was calculated by measuring shoulder alignment relative to the pelvis.

Locomotion

Locomotion was assessed through traditional gait parameters, FLEs, and upper body coordination. Six walking trials (three departures and three returns) were analyzed per participant. In total, an 18 m walkway was assessed.

We hypothesized that PwP would exhibit lower TO and HS angles (reduced ankle dorsiflexion) and lower foot height than controls. Figure 3 illustrates the TO and HS angles during the gait cycle.

**Figure 3 sensors-25-05999-f003:**
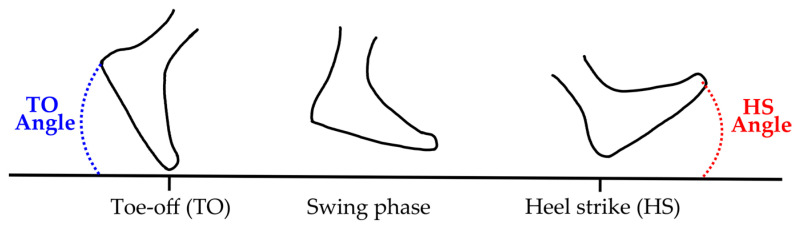
Schematic representation of Toe-off and Heel-strike tasks during the gait cycle.

We predicted PwP would show higher asymmetry, longer double support, shorter stride length, wider stride width, longer stride times, lower stride velocities, and higher variability percentages compared to controls.

Upper body dynamics were assessed by calculating arm swing displacement and angular velocities of three segments: shoulder-elbow, elbow-wrist, and shoulder-wrist.

We calculated an arm asymmetry factor (ASA), defined as the percentage of deviation from perfect symmetry of the shoulder-wrist segment. Based on Lewek et al. (2009) and Zifchock et al. (2008) [118,119], we determined the dominant arm by comparing total arm-swing values for the left and right shoulder-wrist segments, with the dominant arm being the one with the largest swing. The asymmetry factor was calculated as the normalized angle ratio between the dominant and non-dominant arms expressed as an absolute percentage.(2)$$\mathrm{ASA}=\frac{\left(45{}^{\circ}-\arctan\left(Total\;arm\;swing_{Dominant}-Total\;arm\;swing_{Non-dominant}\right)\right)}{90{}^{\circ}}\times 100\%$$

An ASA of 0% indicates perfect symmetry, while a factor of 100% indicates the arm swings are equal and opposite in magnitude [119]. We expected PwP to show reduced arm swing, lower arm segment velocities, and higher ASA than controls.

Freezing

FLE, defined as periods in which walking speed drops below 10% of the baseline speed [82], was identified by comparing the walking speed during the task with the baseline speed, obtained from TUG trials. Walking speed was calculated using the pelvis midpoint displacement as the average of the departure and return walking tasks. The absolute value of the return speed was used, as it is negative due to the trajectory. FLEs were identified using the baseline speed threshold. The duration of each event was measured. FLEs during the FMA-P protocol and the additional locomotion task were calculated.

#### 2.3.8. Performance Score

Two experienced researchers in movement assessment evaluated the ten items and four descriptions of the Performance Score independently of each other. Differences in the evaluation of the tasks were discussed and analyzed by the researchers. The mean score from three trials (FMA-P) was calculated for each item. Final scores were calculated as the sum of all items.

#### 2.3.9. Statistical Analysis: Analysis of Covariance for Group and Protocol Differences

Initially, a MANCOVA analysis was considered; however, our sample size and the number of dependent variables were not ideal for this type of statistical analysis. Instead, a 2 (Group: PwP vs. Control) × 2 (Protocol: TUG vs. FMA-P) ANCOVA was applied to each biomechanical parameter associated with each task.

This approach allowed us to characterize both group-level differences in biomechanical strategy (i.e., how movement unfolded) and how these differences are modulated by testing protocol (TUG vs. FMA-P).

Each biomechanical parameter was analyzed separately as a dependent variable, aligned with the best biomechanical practice. The same set of parameters was examined where possible, and thus multiple comparison corrections were applied to control for inflated Type I error rates.

For within-protocol group comparisons (e.g., Control vs. PwP on TUG), within-group protocol comparisons (e.g., PwP: TUG vs. FMA-P), and for Group × Protocol interaction, the alpha level (α) was adjusted based on the number of parameters analyzed. For example, with six parameters, the corrected α = 0.05/6 = 0.0083.

The number of parameters considered for the Bonferroni corrections was defined by parameter type, such as spatiotemporal (e.g., duration), kinetic (e.g., COP), and upper/lower kinematics (e.g., trunk inclination, stride velocity). In the spatiotemporal category, only one parameter was compared across groups and protocols, and no Bonferroni correction was applied to this parameter.The Shapiro–Wilk test was applied to assess the normality of our data. Sphericity was assessed using Mauchly’s test. Data with sphericity assumption violation was corrected using the Greenhouse-Geiser method. To control observed and unobserved differences in our sample, age, sex, and body height as fixed effects were included in all analyses. Foot length was added as a covariate for biomechanical parameters such as TO angle, HS angle, and foot height comparisons.

### 2.4. Study 2: Application of the FMA-P to Measure Functional Mobility Outcomes of a Music and Movement-Based Intervention

The overarching research question of Study 2 was to assess the sensitivity of the FMA-P in detecting changes in functional mobility in PwP following a music- and movement-based intervention. This study aimed to determine whether the FMA-P can serve as a reliable outcome measure to evaluate motor improvements across time in response to a therapeutically grounded program.

Based on this research question, the following hypotheses were formulated:

**H1:** 
*The FMA-P will detect significant improvements in functional mobility from pre- to post-intervention in PwP.*


**H2:** 
*Improvements will be most evident in FMA-P parameters related to task coordination and transitions between movement phases.*


**H3:** 
*The multidimensional nature of the FMA-P (integrating kinematic and spatiotemporal data) will allow for a more nuanced detection of functional changes compared to conventional outcome measures.*


#### 2.4.1. Study Design

The data used herein is a sub-sample of a larger study evaluating the efficacy of an intervention, co-developed with PwP, that uses music and movement to improve functional mobility and quality of life for PwP [67]. In total, three trials were completed in the UK and Switzerland (*in preparation*). This is a sub-sample of the first group that completed the intervention in the UK, collected between 1 May 2023 and 14 August 2023. The intervention took place once per week over 12 consecutive weeks, with each session lasting 90 min. Although each session of the intervention had a different weekly theme (e.g., Marching Music, Music from Africa or Latin America), the structure was based on the same framework of eight sections related to the therapeutic use of music in Parkinson’s rehabilitation.

The sections included a warm up (seated and standing exercises to music), active rhythmic engagement (learning to find, feel, and play the main pulse of the music), sharing of meaningful songs (group bonding), line dancing (a sequence of movements incorporating exercises related to functional mobility), a discussion section (exploring music in relation to under-researched areas of PD symptoms such as difficulties with sleeping), an informational talk on music of the themed area, a section exploring the music and the dance of the weekly theme, and a final restoration section (i.e., stretching accompanied by music of the weekly theme). Further information on the protocol can be found on the Open Science Framework and as documented in Rose et al., 2025 [67]. Participants in study 2 were assessed through biomechanical motion capture analyses using the FMA-P protocol at three time points—Baseline, Pre-, and Post-intervention—with three measurements taken at each time point, as detailed in Section 2.3. Of the 12 participants included in Study 2, five had previously participated in Study 1, while seven were newly included to allow for time-tracked analyses. All participants were assessed under the same experimental conditions and recruited based on the same inclusion criteria as Study 1, ensuring comparable clinical characteristics at a group level.

#### 2.4.2. Statistical Analysis: RM-ANOVA of Intervention Phases

An RM-ANOVA was conducted to compare the parameters defined in Study 1 (see Table 1) across the three phases of the intervention: Baseline, Pre-, and Post-intervention. Normality and sphericity of our data were assessed. Post hoc pairwise comparisons between intervention phases were implemented using the Bonferroni correction.

## 3. Results

Basic demographic characteristics for all groups are provided in Table 2. After that, the results are grouped according to study.

### 3.1. Results of Study 1

#### 3.1.1. Complementary Clinical Assessments

In Study 1, the Parkinson’s group were in the early stages of PD, according to the Hoehn & Yahr (H&Y; Mean = 1.8, SD = 0.6, Range 1.0–2.5) and presented a wide range of MDS-UPDRS III scores (Mean = 33.7, SD = 12.1, Range 11–51).

A significant difference between groups was found: the PwP were older than the controls, F(2) = 4.43, *p* = 0.025, η_p_^2^ = 0.297.

As expected, a significant difference was found between groups for the Performance Score, whereby the PwP (Mean= 8.4, Range 3.9–17.3) scored higher (worse performance) than the control group (Mean = 2.5; Range 0.9–4.99), with age and sex considered as covariates (F(2) = 5.83, *p* = 0.003, η_p_^2^ = 0.551).

No differences were found between groups for the WHO−5 measure of wellbeing, after adjustments for age and sex (PwP: Mean = 81.8%, Range 64–92; Controls: Mean = 81.8%, Range 44–100; F(3) = 0.63, *p* = 0.606, η_p_^2^ = 0.0.086).

#### 3.1.2. Clinical Measures

For analysis, mean times from three stopwatch-timed trials of the FMA-P and TUG were used, providing data consistent with clinical practice.

As shown in Table 3, the PwP took significantly longer to complete the FMA-P compared to controls, F(1, 22) = 8.16, *p* = 0.009, η_p_^2^ = 0.27, but not for the TUG (*p* = 0.057).

#### 3.1.3. Biomechanical Analysis

This section presents the biomechanical analysis of motion and gait mat data on comparable tasks derived from the TUG and FMA-P (i.e., Sit-to-stand, Turning, and Stand-to-sit). A 2 (Group) × 2 (Protocol) ANCOVA was applied at the level of the biomechanical parameters associated with each task.

Prior to analysis, potential fatigue effects were examined across the three TUG trials performed during our experimental sessions in a selection of 12 participants in each group [125,126]. Intra-class correlation (ICC) analysis with a one-way random-effects model values indicated an ICC(1, 1) of 0.98 (95% CI [0.95, 0.99]) for individual trials and an ICC (1, 3) of 0.99 (95% CI [0.98, 0.99]) for the mean of the three trials. The ICC was statistically significant (F(23, 48) = 125.54, *p* < 0.001), suggesting consistent performance across repetitions and thus confirming the absence of fatigue effects. Given this stability, to reduce the demand on resources that would be required within the research team to label all three TUG trials, we selected the second of the three TUG trials for further analysis for between-groups analyses. However, as the FMA-P is a newly developed measure, the mean of three trials completed by each participant within a single experimental session was used for each of the FMA-P parameters.

Significant results are presented in Table 4; complete results are available in Table A1 and Table A3.

Sit-to-stand

When comparing groups on the Sit-to-stand transition, PwP took significantly longer for the FMA-P protocol than controls (F(1, 52) = 6.62, *p* = 0.012, η_p_^2^ = 0.09), but there was no difference between groups for this aspect of the task on the TUG (*p* = 0.077).

A Bonferroni correction for the group comparison of seven upper-body kinematic parameters was applied, adjusting the significance threshold to α = 0.007, and for two COP parameters to α = 0.025.

The result of a 2 (Group: PwP vs. Healthy Control) × 2 (Protocol: TUG vs. FMA-P) ANCOVA, controlling for age, sex, and height, revealed a significant main effect of Group on peak anterior–posterior (AP) trunk inclination (F(1, 41) = 8.82, *p* = 0.006, η_p_^2^ = 0.17). As illustrated in Figure 4, PwP inclined their trunk further toward the ground (AP direction) than control participants.

The analysis also revealed a significant main effect of Group on the dynamics of trunk movement. Specifically, PwP exhibited lower trunk jerk, standing up with a smoother, more controlled movement than controls (F(1, 40) = 10.22, *p* = 0.003, η_p_^2^ = 0.21).

A main effect of Protocol in the displacement of the COP in the ML direction was found. Both groups exhibited a higher balance shift sideways during TUG than during FMA-P (F(1, 38) = 10.9, *p* = 0.002, η_p_^2^ = 0.22).

Turning

A main effect of Protocol for this task duration (F(1, 36) = 5.41, *p* = 0.026, η_p_^2^= 0.13) was found; both groups took longer to complete the turn on the TUG compared to FMA-P.

The Bonferroni corrections for the group comparison of eleven upper-body kinematic parameters resulted in a significance threshold equal to α = 0.005. For two gait parameters (HS and TO angles) and two COP parameters, the alpha level was corrected to α = 0.025.

A main effect of the FMA-P Protocol for absolute head yaw (F(1, 32) = 18.16, *p* < 0.001, η_p_^2^ = 0.36), the head yaw angle relative to trunk (F(1, 35) = 14.80, *p* < 0.001, η_p_^2^ = 0.30), and head onset (F(1, 33) = 22.15, *p* < 0.001, η_p_^2^ = 0.40) was observed. During the TUG both groups exhibited a wider head segment rotation (Figure 5a) and separated their heads further from their trunks when turning (Figure 5b). Additionally, it was found that the head segment turned earlier during TUG compared to FMA-P, as shown in Figure 6a.

With respect to gait parameters when turning, a main effect of Group for the HS angle (F(1, 35) = 56.21, *p* = 0.010, η_p_^2^ = 0.18) was demonstrated. As presented in Figure 6b, the toe clearance (i.e., distance between the toes and the ground) for PwP was shorter than that of controls while turning during both protocols.

In terms of COP (balance) control during turning execution, a main effect of Protocol on the COP displacement in the ML direction (F(1, 41) = 92.98, *p* < 0.001, η_p_^2^ = 0.69) was found. As shown in Figure 6c, both groups had higher COP displacement in the ML direction during the FMA-P.

Stand-to-sit

As in the Sit-to-stand task, the significance threshold to α = 0.007 for seven upper-body kinematic parameters, and to α = 0.025 for two COP parameters was adjusted. There were no significant findings for this task at either the group or protocol levels.

Functional reach

All participants were able to pick up the key (sometimes requiring more than one attempt), and the time to complete the key-picking task was similar in both groups. No differences were apparent regarding the upper-or lower-body kinematics for this task.

#### 3.1.4. Additional Tasks

Standing upright

After correcting the significance level for multiple comparisons, no significant differences were found between PwP and Controls on this task.

Locomotion

The significance level to α = 0.003 for nineteen gait parameter comparisons, and to α = 0.013 for the comparison of four upper-body kinematic parameters was corrected.

PwP exhibited a significantly lower TO (F(1, 138) = 11.99, *p* < 0.001, η_p_^2^ = 0.16) and HS angles (F(1, 138) = 11.99, *p* < 0.001, η_p_^2^ = 0.1) compared to control (see Figure 7) during the locomotion task. PwP also walked significantly slower (F(1, 139) = 12.93, *p* < 0.001, η_p_^2^ = 0.09). The PwP group performed the task using shorter stride lengths (F(1, 144) = 9.80, *p* < 0.001, η_p_^2^ = 0.67) and with lower stride velocity (F(1, 144) = 10.39, *p* < 0.001, η_p_^2^ = 0.69).

No significant differences were found between groups on other gait metrics, upper limb angular velocities, and ASA when walking. No significant differences between groups were observed during quiet standing. Further results details are available in Table A4.

Freezing

Our algorithm detected a total of 14 FLEs among PwP with a mean duration of 0.43 s (*SD* = 0.5) during FMA-P.

#### 3.1.5. Performance Score

As presented in Section 3.1.1, PwP had a significantly higher score compared to their control participants (F(1, 16) = 5.83, *p* = 0.003, η_p_^2^ = 0.55).

From a qualitative perspective, it was observed that PwP used unexpected strategies to perform the sit-to-stand task within the FMA-P, such as pushing themselves up from a chair with one or both hands, and/or using a single- or double-foot offset to stand up. Similarly, during the stand-to-sit task, PwP frequently relied on hand support on their knees. For the walking task, PwP often displayed an elbow arm swing with variable amplitudes. During the functional reach task, participants usually required additional steps, multiple attempts, or hand support on their knees. Table A5 provides a detailed overview of the strategies that were used.

### 3.2. Results of Study 2

#### 3.2.1. Clinical Analysis and Performance Score

According to the Hoehn & Yahr staging, PwP presented with mild to moderate symptoms (H&Y; Mean = 1.8, SD = 0.6, Range 1.0–2.5). The mean score for the MDS-UPDRS III at baseline was 42.1 (SD = 10.9, range 20–60).

PwP showed a mean Performance Score of 10.4 at baseline (SD= 5.3, range 2–20). Consistent with observations from Study 1, participants demonstrated a variety of compensatory strategies during the FMA-P tasks, as summarized in Table A5.

Mean time differences of three FMA-P and TUG trials, the mean differences of the MDS-UPDRS III score, and Performance Score for Baseline, Pre-, and Post-intervention were shown in Table A6. No significant changes over time were observed.

#### 3.2.2. Biomechanical Analysis

This section presents the biomechanical analysis of motion capture data on the FMA-P tasks during three intervention phases: Baseline (measurements taken 4–6 weeks before the intervention), Pre (1–2 weeks before the intervention), and Post (measurements taken 1–2 weeks after the intervention). The mean of three trials was used for the comparison of the three experimental phases. Significant results are presented in Table 5; complete results are available in Table A7, Table A8, Table A9 and Table A10.

Sit-to-stand

The RM-ANOVA revealed a main effect of Phase on the trunk acceleration RMS in PwP (F(2, 20) = 4.12, *p* = 0.032, η_p_^2^ = 0.29); however, the post hoc test did not identify significant differences between phases (Figure 8). In contrast to the cross-sectional findings from Study 1, no changes over time were observed in task duration, trunk inclination, or mean trunk jerk.

Turning

A main effect of Phase on turn duration (F(2, 22) = 10.83, *p* = 0.001, η_p_^2^ = 0.50) was found. Post hoc pairwise comparisons indicated no significant difference between the Baseline and Pre measures. A significant difference between the completion times of Pre and Post measurements was revealed, suggesting an effect of the intervention. PwP took less time to complete the turn on Post-intervention compared to the Pre measurements (and also between Baseline and Post measures).

There was a main effect of Phase in pelvis jerk (F(2, 18) = 4.31, *p* = 0.030, η_p_^2^ = 0.30). After correction, a marginal difference (*p* = 0.046) between Baseline and Post trunk jerk was found. PwP turned their pelvis more suddenly during Post compared to Baseline. No significant results were found between Pre and Post intervention.

In addition, an effect of Phase on the upper body turning strategies was observed. Specifically, on the turning angles of the head (F(2, 22) = 21.6, *p* < 0.001, η_p_^2^ = 0.66), trunk (F(2, 22) = 21.22, *p* < 0.001, η_p_^2^ = 0.66), and pelvis segments (F(2, 22) = 21.6, *p* < 0.001, η_p_^2^ = 0.66). Results show that PwP exhibited a narrower head, trunk, and pelvis angles during the Post measurements, compared to Baseline and Pre measurement time points as shown in Figure 9.

A main effect of phase on the turn angle of the head relative to the pelvis (F(2, 20) = 3.59, *p* = 0.046, η_p_^2^ = 0.26) was found, but post hoc test did not reveal any significant differences.

Pertaining the trunk alignment during the turn, an effect of phase in the peak inclination in the AP (F(2, 20) = 4.47, *p* = 0.025, η_p_^2^ = 0.31), and ML directions (F(2, 22) = 6.97, *p* = 0.005, η_p_^2^ = 0.39) was presented. The post hoc test only revealed a significant difference in the ML trunk inclination between Pre and Post. PwP exhibited a higher trunk inclination sideways on Post compared to the Pre measurement.

Stand-to-sit

Results indicated the intervention Phase had a significant effect on the maximum pelvis flexion (F(2, 22) = 6.81, *p* = 0.004, η_p_^2^ = 0.38) and knee flexion of PwP (F(2, 22) = 16.49, *p* < 0.001, η_p_^2^ = 0.6). Post hoc test revealed PwP exhibited higher pelvis flexion (body further toward the ground) during Pre measurements compared to Baseline (Figure 10a). Concurrently, PwP had higher knee flexion during Post and Pre measurements compared to Baseline (Figure 10b).

Functional Reach

Results indicated a main effect of the intervention phase in the maximum stance width of PwP when picking up the key from the ground (F(2, 20) = 3.78, *p* = 0.041, η_p_^2^ = 0.27); however, the post hoc test did not reveal significant differences between phases.

## 4. Discussion of Study 1

Study 1 reports the development of a new protocol designed to provide further fine-grained insight into functional mobility in PD than commonly used clinical measures (such as the TUG test). To assess its efficacy, performance on the FMA-P was compared with the TUG [27] using biomechanical parameters, allowing a robust evaluation of differences between PwP and healthy controls, and providing insight into how the paradigm captures aspects of mobility beyond those measured by the TUG. The findings indicate that FMA-P provides additional, goal-oriented insights into functional mobility and motor impairments in early-stage PD.

The results illustrate that while the TUG did not reveal significant differences between groups, the FMA-P, by incorporating a goal-directed functional reach task, did differentiate PwP from controls, with PwP taking longer to complete the goal-directed task. These results suggest that while healthy individuals adapt their motor control or strategies in response to increased task complexity, clinical groups may lack the same flexibility. We suggest, therefore, that the FMA-P helps to magnify between-group differences more so than the traditional metrics completed by the TUG.

For example, biomechanical analysis of sit-to-stand transitions in both the FMA-P and TUG revealed significant differences in upper body kinematics. Specifically, PwP exhibited reduced inclination in the anteroposterior (AP) direction, a pattern which may reflect increased postural instability and the need for anticipatory adjustments to facilitate transition. Indeed, assessment of the Performance Score complements these findings by illustrating that PwP tended to push themselves up from the chair with one or two hands and/or using a foot offset to stand. While previous studies have reported exaggerated pelvis flexion during sit-to-stand in PwP [127], the result of this study indicate that these adaptations are not universal and may be task-specific.

Reduced forward inclination during sit-to-stand suggests that PwP are less reliant on a momentum-driven strategy to initiate this movement [128], while trunk velocities during the completion of the transition remain similar between groups. This findings support Inkster and Eng (2004), who reported that, although sit-to-stand completion times may be comparable between PwP and controls, PwP adapt their movement patterns to compensate for motor deficits [87]. While the RMS of trunk acceleration has previously been proposed as a discriminative metric between PwP and controls, these results suggest that it may not effectively differentiate early-stage PD [123].

Analysis of the turning phase of the FMA-P and TUG further elucidated motor and postural deficits in our PwP cohort. Delayed head onset and head rotation relative to trunk rotation in PwP suggest reduced segmental mobility. Participants also demonstrated significantly reduced rotation angles of the head, trunk, and pelvic segments during the FMA-P protocol. These findings may reflect the increased complexity and planning requirements of the protocol (as the next task was to pick up a key), which may require more frequent or forceful corrections during the movement execution–consistent with a more en-bloc strategy. Additionally, the altered segmental patterns may indicate a reliance on visuomotor or attentional strategies to support action planning for the FMA-P. Paradoxically, while reduced segmental rotation suggests constrained movement, it may also reflect heightened anticipatory control and effort to maintain stability in this more challenging protocol. Notably, this pattern is more pronounced in PwP, despite participants being in the early stages of disease progression, with predominantly mild, bilateral motor symptoms and preserved balance (as indicated by a mean Hoehn & Yahr score of 1.8).

Furthermore, significantly lower mean TO angles in PwP point to altered foot propulsion and landing dynamics, as reported by Schlachetzki et al. (2017) [129,130], while COP displacement during turning was predominantly affected in the ML direction—an adaptation that may have implications for targeted rehabilitation. Reduced TO angles across both protocols indicate a more conservative or rigid stepping strategy. This may reflect impaired motor flexibility or anticipatory control. Interestingly, control participants also demonstrated a trend toward a reduction in TO angles during the FMA-P task, perhaps reflective of the increased task demands.

However, the functional reach task introduced in the FMA-P did not reveal any significant group differences. This may be attributed to the relatively homogenous nature of the groups, and/or the relatively low level of symptom severity in the PwP group. Nevertheless, it is notable that differences were found between groups for the other functional mobility tasks embedded in the FMA-P, suggesting that whilst the task increases the demand, overall, it is not the task itself that is difficult. In line with guidance about patient and public involvement in research, we conducted trials of several items, including picking up and placing a tissue box from a table to above shoulder height, and picking up a heavy bottle. Still, the PwP we worked with found picking up and placing a key to be the most ‘realistic’ task. Future studies could consider the impact of alternatives for this aspect of the sequence.

Similarly, after correcting for multiple comparisons, no significant differences were observed between PwP and control participants during the quiet standing task. Although measures such as postural sway (COP displacement) and jerk have been suggested as early indicators of balance impairment in individuals with mid-stage PD [128,131,132], they may not reflect balance challenges in early-stage PwP, as also stated by previous studies [133,134]. Further examination should be considered with a more diverse sample of PwP.

The locomotion task results indicated significant gait disturbances in the PwP group. Specifically, PwP demonstrated significantly lower toe-off (TO) and heel-strike (HS) angles, as well as reduced overall foot height compared to controls, consistent with previous studies and reflecting altered foot mechanics in PD [130]. Additionally, even at an early stage of PD, these results demonstrated that PwP had a reduced stride length and slower walking velocity, suggesting reduced propulsion [88]. The prolonged double support time of PwP highlights the need for stability, reflecting compensatory mechanisms for balance preservation.

No significant differences in arm swing velocity and asymmetry were identified between groups, which may reflect variability in arm swing dynamics in early and mild PD. These results were also reflected in the qualitative results of the Performance Score, with individuals tending to perform the arm swing from the elbow, but with very different individual characteristics (i.e., amplitude and asymmetry). This approach supports the individualization of rehabilitation programs, which is an important goal of Parkinson’s care.

Notably, 14 freezing-like episodes were observed in PwP during the FMA-P where participants walked through the doorway. In contrast, no FLEs occurred when PwP walked without a doorway, implying environmental features like doorframes could influence freezing behavior, as reported by previous studies [82].

Despite the insights gained in understanding qualitative differences in functional mobility through the integration of motion capture and pressure-sensitive gait mat technologies, the study has several limitations that need to be addressed. Firstly, findings are limited as the sample of PwP was in the early stages of the disease. Nevertheless, even at this early stage, the paradigm enabled identification of functional mobility impairments in PwP compared to healthy individuals over and above what was found even with motion capture analysis of the TUG, suggesting the FMA-P offers additional insights into the quality of functional mobility in Parkinson’s that has implications for both intervention studies and rehabilitation strategies.

## 5. Discussion of Study 2

Study 2 aimed to evaluate changes in functional mobility in PwP across three intervention phases using the FMA-P. While traditional clinical metrics such as the UPDRS, TUG, and task duration did not reflect significant improvements, motion capture analysis of the FMA-P captured meaningful biomechanical adaptations, particularly in tasks involving turning, stand-to-sit transitions, and functional reach. These findings underscore the FMA-P’s potential for detecting subtle rehabilitation-related changes in motor performance.

In contrast to Study 1, where sit-to-stand transitions revealed between-group differences in forward trunk inclination and compensatory strategies, Study 2 identified a significant effect of intervention phase on trunk acceleration (RMS) during the same task. Although post hoc comparisons did not yield significant pairwise differences, the observed results align with previous research suggesting that higher trunk acceleration during sit-to-stand transitions is a relevant indicator of improved postural control in PwP [124].

The stand-to-sit task revealed phase-related changes in joint flexion strategies. PwP demonstrated increased maximum pelvis and knee flexion from Baseline to Post-intervention. These results may reflect improved eccentric control and a more deliberate, confident approach to sitting, although higher angles could also indicate a more cautious descent strategy [87].

Turning, which in Study 1 differentiated PwP from controls through reduced segmental rotation and delayed head onset, emerged in Study 2 as the most sensitive task to intervention effects. Following the intervention, PwP demonstrated significantly shorter turns, pointing to improved motor planning and execution. In addition, an increase in pelvis jerk post-intervention may reflect more dynamic and confident turning behavior, consistent with a shift in movement strategy. As previous research has shown a link between pelvis jerk and fall risk in PD, it serves as a relevant marker for assessing dynamic balance and movement control [135].

Segmental yaw rotation angles were reduced post-intervention. Rather than suggesting rigidity, this likely reflects more efficient axial coordination, enabling faster and more controlled turning, as reported by previous studies [91].

Additionally, significant effects were found in trunk inclination: both AP and ML peak angles changed across phases, with ML inclination significantly increased from Pre to Post. This suggests greater lateral trunk engagement, possibly reflecting improved balance and a more proactive turning strategy.

In contrast to Study 1, where functional reach did not differentiate between groups, Study 2 demonstrated a significant phase effect on stance width during the reaching task. A wider base of support from Pre to Post-intervention suggests improved anticipatory postural adjustments [136]. This may represent a compensatory response to perceived instability during forward reach, indicating subtle but meaningful changes in movement strategy following intervention.

Notably, while Study 1 identified freezing-like episodes in response to environmental cues (e.g., doorways), Study 2 did not observe consistent changes in freezing behavior across phases. In line with previous research, this may indicate that freezing episodes are not easily influenced during controlled assessments or may not translate into a reduced number of FLEs in short-term standardized tasks [137].

Although no significant improvements were detected in the Performance Score over time, the qualitative data indicate that PwP utilized various adaptive strategies to complete the tasks. These individualized compensations may reflect attempts to maintain function despite underlying motor impairments, underscoring the need for quantitative biomechanical analyses to capture subtle motor changes.

Overall, the findings of Study 2 build upon the results of Study 1 by showing that the FMA-P is not only sensitive to between-group differences but also to within-subject changes in functional mobility over time. Whereas conventional clinical measures failed to detect intervention-related effects, the FMA-P identified significant improvements in turning speed, pelvis jerk, and trunk movements related to stance and goal-directed reach-grasp actions. These changes indicate enhanced dynamic and confident movement strategies, which are critical for reducing fall risk and maintaining functional independence in PwP.

## 6. Implications, Strengths, and Limitations

The TUG framework raises important questions about the interplay between automatic and goal-directed movement control. Movements such as rising from a chair or turning may typically rely on automatic, habitual processes; however, these actions also demand cognitive resources and may engage goal-directed control mechanisms, especially in individuals with PD [75,138]. The extent to which early-stage PD impacts habitual movement, potentially increasing reliance on goal-directed strategies, is not yet fully understood [76]. Moreover, the sensitivity of the TUG to detect subtle impairments in early PD is limited [29,34], as it primarily assesses overall task completion time rather than detailed movement quality. Therefore, to overcome this limitation, we assessed performance through a biomechanical analysis of movement execution, which complements clinical time-based assessments and provides insight into underlying neurological mechanisms, guiding targeted interventions.

The novel FMA-P protocol directly addresses these challenges by combining goal-directed tasks with high-resolution biomechanical assessments to provide a more fine-grained, multidimensional perspective on motor performance in PD. The protocol was well tolerated by participants, and the data obtained revealed functional impairments and movement adaptations not detectable by conventional tools. For example, even in early-stage PD, participants displayed compensatory movement strategies such as altered trunk and pelvis motion or reliance on hand support, likely reflecting underlying motor control deficits that are masked in traditional summary scores.

A key strength of our approach lies in its ability to capture both quantitative and qualitative aspects of motor behavior. Motion capture and pressure-sensitive gait analysis allowed us to detect subtle spatiotemporal and segmental kinematic differences, offering insights into compensatory strategies and adaptive responses to task demands. The qualitative data, such as variations in arm swing pattern or sit-to-stand technique, highlighted substantial inter-individual differences in motor behavior. These differences suggest that individualized analysis of movement patterns could be valuable for tailoring interventions in future studies [49,55].

Certain limitations should be acknowledged. The relatively small sample size and focus on early-stage PD may limit the generalizability of findings to broader clinical populations [139]. Additionally, in Study 1, PwP were significantly older than the healthy control group. Given that age itself can influence balance and gait, this age difference may have contributed to observed between-group differences [139]). Nevertheless, Study 1 was intended as a pilot investigation to evaluate the feasibility and sensitivity of the FMA-P. Importantly, Study 2 employed a within-subject design, thereby eliminating age-related confounds and confirming the FMA-P’s sensitivity to intervention-related changes in functional mobility. To extend these findings, further intervention trials have been conducted in the UK and Switzerland [140], where the clinical utility of the FMA-P and the Performance Score will be further profiled across more diverse and representative cohorts (in preparation).

This study employed motion capture technology to robustly quantify whole-body kinematics, allowing for a comprehensive evaluation of the FMA-P in relation to the clinical standard, the TUG [44,141]. These data provide a high-resolution reference that will inform the identification of key features, which can subsequently be trialed using a combination of the FMA-P and low-cost inertial sensors to facilitate broader clinical implementation [45]. The FMA-P protocol was well received by participants, who reported minimal burden during assessment, and the comprehensive yet streamlined data-analysis approach justified the additional time required, supporting its continued use in future studies. Nonetheless, biomechanical assessments with optical motion capture are resource-intensive and may be impractical for routine clinical application [142]. A further limitation of motion capture is the occasional occlusion of markers. It was selected for its high spatial resolution and ability to capture detailed whole-body kinematics; this trade-off was accepted in the present study to establish a robust framework for profiling subtle whole-body dynamics. Future work should aim to validate features with wearable technologies to enhance scalability and real-world applicability [44].

Finally, the considerable variability in movement strategies observed among participants highlights the importance of individualized assessment and intervention planning, which this approach enables by capturing both quantitative and qualitative motor features.

## 7. Conclusions

The newly developed measure of functional mobility (FMA-P) has been designed for, and with, PwP to reflect the challenges they face in everyday movement sequences. Standard clinical-level measures, such as task completion in the TUG, provide a general mobility assessment but do not capture subtle motor deficits.

Results demonstrate no group differences or intervention effects using the gold-standard measure of the TUG and the MDS-UPDRS. Yet, when using a goal-directed task, subtle variations are revealed, which may reflect the increased complexity of the FMA-P. The FMA-P enables a fine-grained quantification of the quality of tasks such as standing up, walking through a doorframe, turning, picking up and placing an object, and sitting down within one short sequence. The integrated motion capture and pressure-sensitive gait mat system provides multi-dimensional and time-tracked data on these tasks, can improve the understanding of clinical measures, and provides insight into changes in PwP’s various motor symptoms over time. These findings provide objective and precise information about various qualitative aspects of functional mobility to improve the evidence base of outcome variables. Future research will have to address the potential use of the FMA-P within a broader clinical population to assess individual and specific functional motor impairments. The knowledge generated by this method may help inform intervention programs targeting specific tasks in daily life activities, which are aimed at effectively counteracting disease progression and improving quality of life.

## Figures and Tables

**Figure 1 sensors-25-05999-f001:**
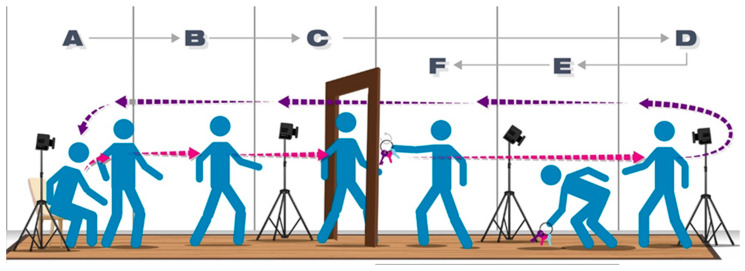
Diagram of the Functional Mobility Assessment in Parkinson’s (FMA-P) sequence. The FMA-P sequence is composed of A. Sitting to standing and standing to sitting, B. Walking forward (3 m. distance), C. Walking through a visual obstacle, D. Turning, E. Bending to pick up an object from the ground, and grasp it, F. Placing the object at the height of the shoulder.

**Figure 2 sensors-25-05999-f002:**
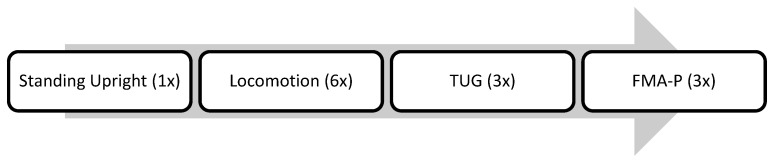
Participants were first instructed to stand upright for 10 s (in the middle of the gait mat). Next, they were asked to walk forwards along the length of the gait mat 6 times, turning on the mat before recommencing the next lap until all six laps were completed. Participants were then asked to perform the TUG three times, each time following the researcher’s instructions, before finally executing the FMA-P three times, also starting on the instructions of the researcher. Short breaks were provided between tasks to minimize fatigue and ensure consistent performance. All recordings took place on the mat within the motion capture suite, as shown in Figure 1.

**Figure 4 sensors-25-05999-f004:**
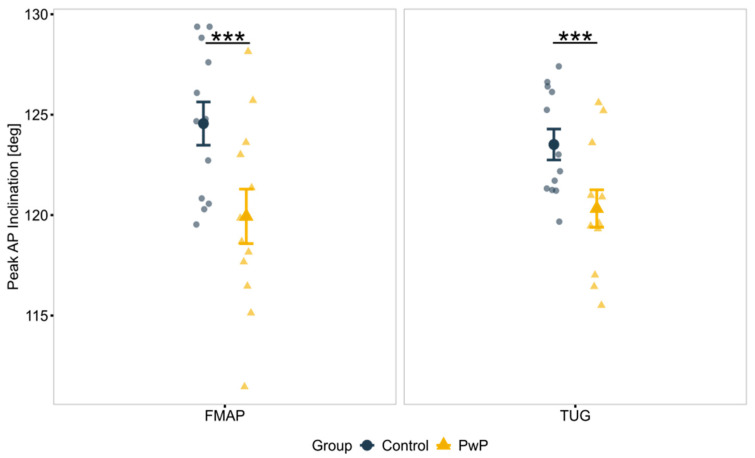
Plot showing mean peak trunk inclination in the anterior–posterior (AP) direction for each group (PwP vs. Control) across two protocols (FMA-P and TUG). Individual data points are overlaid around each group’s mean. Error bars represent the standard error of the mean (SEM). A significant main effect of Group was found (*p* < 0.001), with PwP showing greater forward trunk inclination during sit-to-stand compared to controls. *** *p* < 0.001.

**Figure 5 sensors-25-05999-f005:**
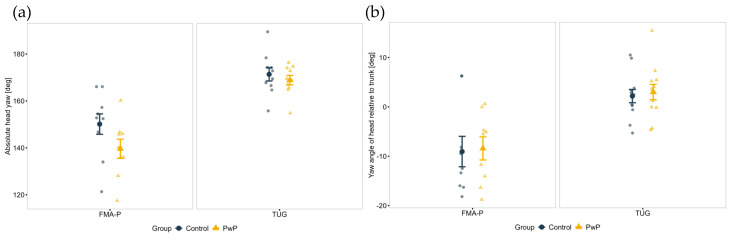
Plots showing parameters of the turning task for each group (PwP vs. Control) across two protocols (FMA−P and TUG). (**a**) Absolute head yaw: both groups exhibited a wider head segment rotation during TUG (*p* < 0.001). (**b**) Yaw angle of the head relative to the trunk; both groups separated their head further from their trunk when turning during TUG.

**Figure 6 sensors-25-05999-f006:**
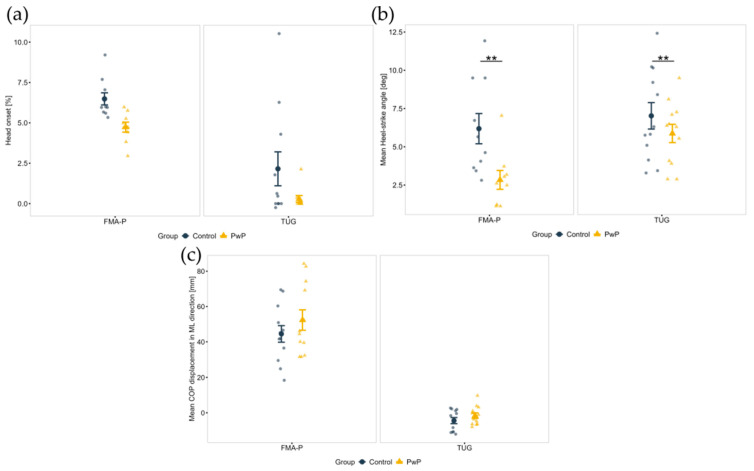
(**a**) The mean head turn-onset for each group (PwP vs. Control) across two protocols (FMA-P and TUG). Individual data points are overlaid around each group’s mean. Error bars represent the standard error of the mean (SEM). A significant main effect of Protocol was found (*p* < 0.001). Both groups had earlier head turns during the TUG. (**b**) Heel-strike angle during the turning task for each group (PwP vs. Control) across two protocols (FMA-P and TUG. A significant main effect of Group was found (*p* =.010). PwP toe clearance was reduced compared to controls during both protocols. (**c**) COP displacement in the ML direction during the turning task for each group (PwP vs. Control) across two protocols (FMA-P and TUG). A significant main effect of Protocol was found (*p* < 0.001). Both groups exhibited higher COP mediolateral displacement during the FMA-P. ** *p* < 0.01.

**Figure 7 sensors-25-05999-f007:**
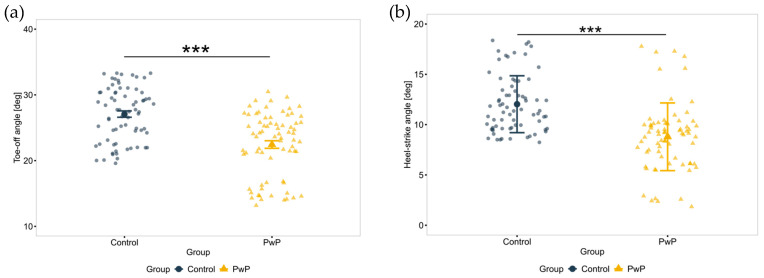
Plots showing the mean (**a**) Toe-off (TO) and (**b**) Heel-strike (HS) angles are displayed for each group of participants. The TO angle is the highest angle between the heel and the ground at the beginning of the stride, while the HS angle is the highest angle between the toe and the floor at the end of the stride. Statistical differences between groups were found with an ANCOVA analysis and significance level correction with the Bonferroni method; detailed results are shown in Table A4. PwP, People with Parkinson’s. *** *p* < 0.001.

**Figure 8 sensors-25-05999-f008:**
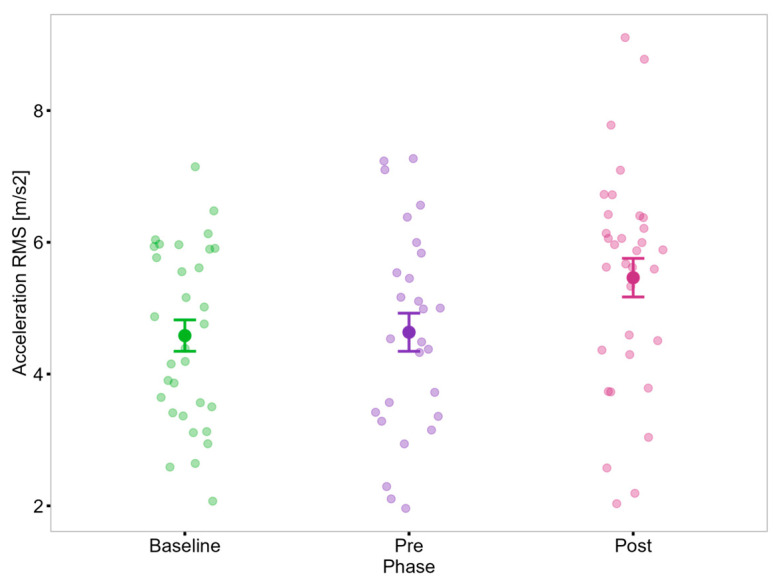
Plot showing the trunk acceleration RMS during the sit-to-stand task for each experimental phase (Baseline, Pre, and Post) during Sit-to-stand. Individual data points are overlaid around each time point. Error bars represent the standard error of the mean (SEM). A significant main effect of the experimental phase was found (*p* = 0.032); however, post hoc tests did not reveal significant differences between phases.

**Figure 9 sensors-25-05999-f009:**
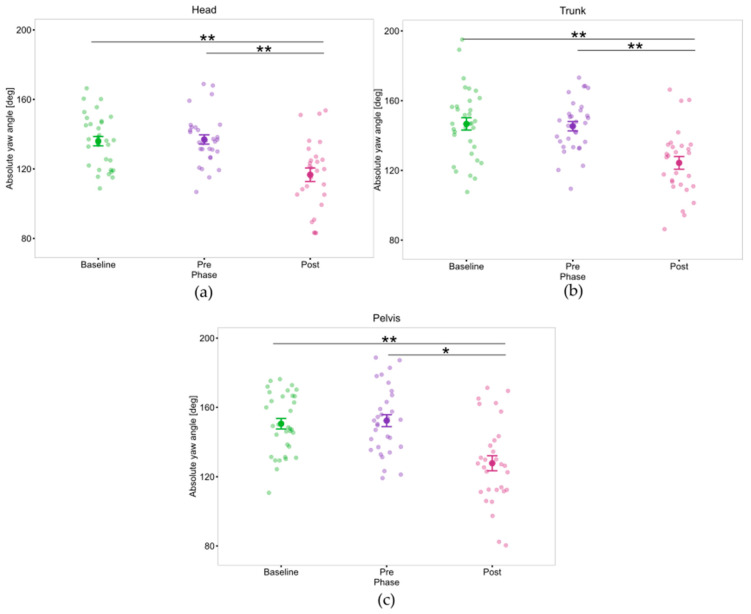
Plot showing the body segment’s rotation angles of the (**a**) Head, (**b**) Trunk, and (**c**) Pelvis for each experimental phase (Baseline, Pre, and Post) during Turning. Individual data points are overlaid around each time point. Error bars represent the standard error of the mean (SEM). Post hoc tests revealed PwP’s post-rotation angles were reduced compared to those of Baseline and Pre measurements. * *p* < 0.05, ** *p* < 0.01.

**Figure 10 sensors-25-05999-f010:**
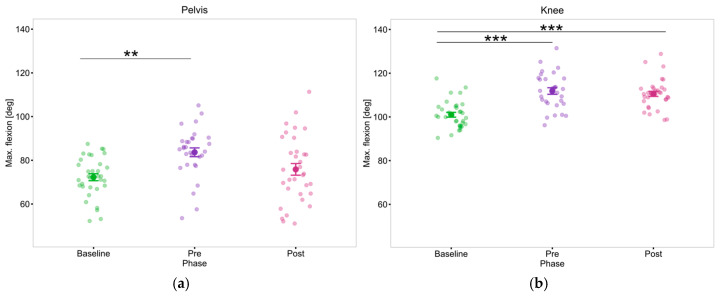
Boxplot showing the maximum flexion of (**a**) Pelvis and (**b**) Knee for each experimental phase (Baseline, Pre, and Post) during Stand-to-sit. Individual data points are overlaid around each time point. Error bars represent the standard error of the mean (SEM). Results show PwP had higher pelvis flexion during Pre measures compared to Baseline and higher knee flexion during Pre and Post compared to Baseline. ** *p* < 0.01, *** *p* < 0.001.

**Table 1 sensors-25-05999-t001:** Overview of biomechanical and gait metrics analyzed in this study.

Parameter	Tasks	Description	Predictor/Expected Difference	References
Max trunk inclination (AP, ML)	Sit-to-Stand/Stand-to-Sit; Turning	Max. angle between trunk and vertical axis with respect to AP/ML planes.	Larger values may indicate compensatory leaning. Excessive tilt expected in PwP, especially in the ML axis during turning.	[94]
Peak trunk velocity (AP)	Sit-to-Stand/Stand-to-Sit	Derivative of trunk displacement.	Higher values indicate momentum-driven strategies; lower values may indicate rigid movement; lower values expected in PwP.	[94]
RMS trunk acceleration (AP)	Sit-to-Stand/Stand-to-Sit; Standing Upright	Measure of anticipatory postural adjustments.	Lower values in PwP indicate reduced anticipatory postural adjustments.	[94]
Mean/Trunk & Pelvis jerk	Sit-to-Stand/Stand-to-Sit; Turning; Standing Upright	Derivative of trunk acceleration, measures the smoothness of movement.	Higher values in PwP may indicate less fluid/postural control. During Turning, reduced jerk reflects less dynamic compensatory movement. In Standing Upright, indicator of movement smoothness.	[73,94,108]
Peak pelvis flexion	Sit-to-Stand/Stand-to-Sit; Functional Reach	Angle between trunk–pelvis and trunk–knees vectors.	During Sit-to-stand/Stand-to-sit greater flexion in PwP reflects reliance on pelvis flexion to stabilize COM during transition. In Functional Reach, reduced pelvis flexion may indicate compensatory while bending down.	[103,105]
Knee flexion	Sit-to-Stand/Stand-to-Sit; Functional Reach	Angle between knee–pelvis and knee–heel.	Greater knee flexion values in PwP suggest a compensatory strategy for stability during sit-to-stand and stand-to-sit, and similarly, greater flexion during Functional Reach reflects a compensatory strategy while bending down.	[87,104,105]
COP displacement/sway (AP, ML)	Sit-to-Stand/Stand-to-Sit; Turning; Functional Reach; Standing Upright	Postural stability indicator. Obtained from the PKMAS system.	Greater displacement in PwP reflects impaired balance/anticipatory postural control.	[8,73,107]
Absolute yaw rotation angle	Turning	Segment rotation (head, trunk, pelvis) around the vertical axis.	Reduced angles indicate in-bloc turning strategy. Higher angles suggest a craniocaudal turning strategy.	[91]
Relative yaw angle	Turning	Relative rotation between head–trunk, trunk–pelvis, and head–pelvis.	Indicator of flexibility and coordination between adjacent body segments. Smaller differences in PwP suggest less coordinated turning strategy.	[91,92]
Onset of rotation (% gait cycle)	Turning	Percentage of gait cycle for head, trunk, pelvis normalized to the first stride of turn.	Additional indicator to determine turning strategy. Lower onset values = earlier rotation; differences indicate altered coordination in PwP.	[91,92]
Arm alignment angle	Functional Reach	Shoulder–wrist segment alignment with respect to the sagittal plane.	Indicator of upper body control while bending down. Values close to 90° indicate a diagonal reach further from the body; PwP expected to show higher variability.	[75]
Stance stability: AP distance, stance width, feet aperture angle	Functional Reach	Frontal and mediolateral distance between feet. Angle between the long axes of the feet in the transverse plane.	Larger distances/angles may indicate a compensatory strategy to maintain COM stability while bending down.	[105,106]
Toe-off (TO) angle	Functional Reach; Locomotion	Maximum heel-ground angle (see Figure 3).	Reduced values may indicate impaired dorsiflexion of the ankle joint in PwP.	[88,104]
Heel-strike (HS) angle	Locomotion	Max toe-ground angle (see Figure 3).	Reduced values may indicate shuffling.	[88,104]
Foot height, (minimum foot clearance, mFC)	Locomotion	Shortest vertical foot–ground distance.	Reduced values suggest impaired foot clearance in PwP.	[88,104]
Spatiotemporal gait parameters	Locomotion	Stride length, width, velocity, time, double support, asymmetry, variability.	PwP expected to show higher asymmetry, longer double support, shorter strides, wider stance, slower velocity, and higher variability.	[29]
Arm swing displacement & angular velocities	Locomotion	Arm segment kinematics (shoulder–elbow, elbow–wrist, shoulder–wrist). Calculated using markers coordinated projected onto the sagittal plane, with the pelvis midpoint as the reference point.	Reduced magnitude indicating impaired upper-limb contribution to gait. Lower velocities in PwP, reflecting bradykinesia and reduced rhythmic coordination.	[34]
Arm Swing Asymmetry (ASA)	Locomotion	Asymmetry percentage between dominant and non-dominant arm swing (see Equation (2)).	High asymmetry percentages suggest an impaired arm coordination during gait.	[109,110]
Freezing-like events (FLE)	Locomotion Gait during FMA-P	Based on pelvis midpoint displacement. Detected when the velocity decreases to less than 10% of a baseline. Baseline speed was determined from TUG. FLE duration was calculated considering a minimum duration threshold of 0.5 s.	Higher frequency and longer duration in PwP, reflecting impaired gait initiation and motor blocks.	[118,119]

AP, Anteroposterior direction; ML, Mediolateral direction; RMS, Root-mean-square; COP, Center of pressure.

**Table 2 sensors-25-05999-t002:** Group Demographics for Study 1 and Study 2.

Parameter	Study 1 PwP, *n* = 12		Study 1 Control, *n* = 12		Study 2, *n* = 12	
	M	SD	M	SD	M	SD
Age (years)	69.5	6.1	61.0	7.3	72.2	7.0
Gender						
Female (%)	58		33		67	
Male (%)	42		67		33	

PwP, People with Parkinson; M, Mean; SD, Standard deviation.

**Table 3 sensors-25-05999-t003:** Comparison of Completion Times for the FMA-P and TUG Protocols for Both Groups.

Measure	Group	
	PwP	Controls
TUG		
Mean completion time (s)	10.4	7.3
SD	3.6	1.0
FMA-P		
Mean completion time (s)	13.4	9.7
SD	4.3	1.4

PwP, People with Parkinson; SD, Standard deviation.

**Table 4 sensors-25-05999-t004:** ANCOVA Results for Group × Protocol Comparison in Study 1.

Task	Parameter	Group	TUG M (SD)	FMA-P M (SD)	Protocol Effect (*p*)	Group Effect (*p*)	Interaction (*p*)
Sit-to-stand	Peak Trunk AP Inclination (deg) ^a^	Control	123.5 (2.7)	124.7 (3.8)	0.498	0.006 **	0.365
		PwP	120.3 (3.2)	119.7 (4.9)			
	Mean trunk jerk (m/s^3^) ^a^	Control	−4.0 (1.6)	−4.2 (1.5)	0.270	0.003 **	0.687
		PwP	−2.0 (1.0)	−2.4 (1.5)			
Turning	Task duration (s)	Control	1.5 (0.3)	1.2 (0.2)	0.064	0.026 *	0.951
		PwP	1.9 (0.5)	1.6 (0.5)			
	Absolute head yaw rotation (deg) ^a^	Control	171.3 (9.0)	150.1 (13.7)	<0.001 ***	0.052	0.245
		PwP	168.8 (6.3)	139.6 (12.3)			
	Head relative to trunk yaw rotation (deg) ^a^	Control	2.2 (4.7)	−9.1 (9.2)	<0.001 ***	0.521	0.848
		PwP	3.0 (5.4)	−8.4 (7.0)			
	Head onset (%) ^a^	Control	2.2 (3.5)	6.5 (1.2)	<0.001 ***	0.218	0.904
		PwP	0.3 (0.7)	4.7 (0.9)			
	Mean heel-strike angle (deg) ^b^	Control	7.0 (3.1)	6.2 (3.1)	0.527	0.007 **	0.209
		PwP	5.9 (2.1)	2.8 (1.8)			
	Mean COP displacement in ML (mm) ^b^	Control	−4.1 (5.9)	44.5 (16.2)	0.002 **	0.386	0.098
		PwP	−1.7 (5.6)	52.3 (19.9)			

PwP, People with Parkinson; M, Mean; SD, Standard deviation; COP, Center of Pressure. AP, Anteroposterior direction; ML, Mediolateral direction. ^a^ Parameters adjusted for sex, age, and body height. ^b^ Parameters adjusted for sex, age, body height, and foot length. * *p* < 0.05, ** *p* < 0.01, *** *p* < 0.001.

**Table 5 sensors-25-05999-t005:** Post hoc results from RM-ANOVA analysis of intervention phases.

Task	Parameter	Comparison	Mean Difference	95% CI	t	*p*
Turning	Task duration (s)	Baseline-Pre	−0.02	[−0.3; 0.2]	0.22	1.0
		Baseline-Post	0.4	[0.1; 0.6]	3.97	0.006 **
		Pre-Post	0.4	[0.1; 0.7]	3.98	0.006 **
	Mean pelvis jerk (m/s^3^)	Baseline-Pre	189.2	[−387.4; 765.8]	0.96	1.0
		Baseline-Post	556.3	[−3.8; 1116.3]	2.91	0.05 *
		Pre-Post	367.0	[−192.1; 926.2]	1.92	0.258
	Absolute head yaw rotation (deg)	Baseline-Pre	0.6	[−5.4; 6.5]	0.27	1.0
		Baseline-Post	25.5	[10.4; 40.6]	4.75	0.002 **
		Pre-Post	24.9	[10.6; 39.2]	4.92	0.001 **
	Absolute trunk yaw rotation (deg)	Baseline-Pre	2.1	[−5.9; 10.1]	0.74	1.0
		Baseline-Post	27.4	[11.9; 42.9]	4.98	0.001 **
		Pre-Post	25.3	[10.5; 40.0]	4.83	0.002 **
	Absolute pelvis yaw rotation (deg)	Baseline-Pre	−2.6	[−16.4; 11.3]	−0.52	1.0
		Baseline-Post	21.0	[6.3; 35.7]	4.03	0.006 **
		Pre-Post	23.6	[5.2; 42.0]	3.61	0.012 *
	Max. trunk ML inclination (deg)	Baseline-Pre	−0.1	[−1.9; 1.7]	−0.12	1.0
		Baseline-Post	2.0	[−0.1; 4.2]	2.68	0.064
		Pre-Post	2.1	[0.7; 3.5]	4.24	0.004 **
Stand-to-sit	Max. pelvis flexion (deg)	Baseline-Pre	−12	[−19.9; −4.2]	−4.32	0.004 **
		Baseline-Post	−3.8	[−12.4; 4.9]	−1.23	0.737
		Pre-Post	8.3	[−3.1; 19.7]	2.05	0.194
	Max. knee flexion (deg)	Baseline-Pre	−11.7	[−16.9; −6.5]	−6.39	<0.001 ***
		Baseline-Post	−9.5	[−14.1; −4.8]	−5.76	<0.001 ***
		Pre-Post	2.2	[−5.7; 10.2]	0.79	1.0
Functional reach	Max. stance width (m)	Baseline-Pre	−23.7	[−53.1; 5.7]	−2.31	0.129
		Baseline-Post	0.8	[−30.3; 31.8]	0.07	1.0
		Pre-Post	24.5	[−2; 51]	2.65	0.073

CI, confidence interval; ML, Mediolateral direction. Note. p-values are Bonferroni-adjusted. * *p* < 0.05, ** *p* < 0.01, *** *p* < 0.001.

## Data Availability

A dataset containing the anonymized data will be made available on the institutional repository (LORY, Lucerne Open Repository).

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
