# Peer review of "A New Methodological Approach Integrating Motion Capture and Pressure-Sensitive Gait Data to Assess Functional Mobility in Parkinson’s Disease: A Two-Phase Study"

_sensors, 2025, doi:10.3390/s25195999_

Round 1
Reviewer 1 Report (New Reviewer)
Comments and Suggestions for Authors
The paper presents a novel measure of functional mobility for people with Parkinson’s disease, aiming to assess daily movement challenges. The authors conducted two studies to compare this new measure with the standard TUG test and demonstrated the potential of their approach to quantify motor deficits using a motion capture system and force plates. The manuscript is well-written, clearly structured, and the methods and experimental protocols are described in detail. The experiments themselves are comprehensive and carefully designed, and the results are thoroughly presented and thoughtfully discussed. Overall, this is a strong paper that makes a valuable contribution. A few clarifications and refinements are recommended.
- Please use title-style capitalization for the paper title in accordance with the journal’s guidelines.
- I recommend including descriptions of the marker placement for the Vicon motion capture system, accompanied by a figure to visualize it, as this would help readers better understand and reproduce the setup.
- In the text, there appears to be a section reference issue: “The TUG was administered prior to the FMA-P but after the additional tasks described in section 2.3.3.” From my reading, the additional tasks were described in Section 2.3.2 rather than 2.3.3. It would be good to carefully check all section references throughout the paper to ensure accuracy.
- Another point relates to the order of the tasks. Why were they assigned in the specific sequence described? Does this order matter for the outcomes? Fatigue effects could arise after certain tasks, which might influence the results or their interpretation, so a clarification would strengthen the methodology section.
- I recommend including a table summarizing the features used for functional mobility analysis as described in Section 2.3.7. This would provide a concise overview and allow readers to quickly understand the basis of the analysis.
- In Table 1, there appears to be an extra “a” after “Age (years)” that should be corrected. Additionally, while the detailed results are valuable, the manuscript currently contains too many figures and tables in the main text. Consider moving some of the results to an appendix or merging figures where possible to make the presentation more concise.
Author Response
Please see the attachment.

Reviewer 2 Report (New Reviewer)
Comments and Suggestions for Authors
This paper focuses on PD assessment. The authors developed the Functional Mobility Assessment for Parkinson's (FMA-P), combining motion capture and gait analysis. A pilot study with 12 PD patients and 12 controls showed FMA-P detected movement impairments, demonstrating postural stability issues in chair rise and turning tasks. A subsequent 12-week intervention on 12 PD patients revealed significant improvements in turning stability and balance with FMA-P.
Therefore, the FMA-P offers in-depth PD movement impairment insights, assessing the previously ignored balance and gait, and identifying yaw rotation in turning as a key marker for targeted rehab.
Besides, the research is with systematic and innovative methodology and high scientific rigor. The reviewer considers it ready for publication at the present form.
Author Response
Dear Reviewer 2,
Thank you very much for your thoughtful and positive evaluation of our work. We greatly appreciate your recognition of the FMA-P as a tool that provides detailed insights into PD-related movement impairments, particularly in assessing balance, gait, and turning stability.
We have carefully addressed the other comments provided by the reviewers, which has notably improved the manuscript. We are grateful for your support and for acknowledging the systematic and innovative methodology of our study. Your feedback encourages us and reinforces the relevance of our research for targeted rehabilitation in Parkinson’s disease.
Reviewer 3 Report (New Reviewer)
Comments and Suggestions for Authors
The authors present a novel method for Parkinson's disease (PD) assessment that exploits motion capture and pressure-based gait analysis to account for functional mobility. The choice of highlighting the two different studies is appreciable.
However, many aspects should be clarified and/or improved to enhance the quality of the study. The main concerns the authors should carefully address are reported point-by-point as follows.
Introduction
- The usage of deep-brain stimulation for the treatment of late-stage PD symptoms (https://doi.org/10.3390/make7030084, https://doi.org/10.1016/j.wneu.2023.07.103), as well as the advantages of functional mobility training with respect to the invasiveness of DBS, can be added for the sake of a more comprehensive background. Moreover, a brief insight about the MRI-based diagnosis of PD could be included as well, since inadequate diagnoses have been mentioned.
- The first details about prevalence should be inserted at the beginning of the Introduction, followed by the ones about dopaminergic cells and those of motor symptoms, which are closer to the rehabilitation realm.
- The paper briefly mentions inertial sensors, but the benefits provided by wearables - e.g., illumination independence, occlusion, and portability (https://www.mdpi.com/1424-8220/25/1/260, https://doi.org/10.1007/978-3-030-01845-0_224, 10.1109/LRA.2019.2928775 ) - should be discussed in such a way to sufficiently justify the choice for a fixed motion capture system within the proposed framework.
- At least one paragraph providing background about music-and-movement-based intervention should be included to support its usefulness in the rehabilitation context.
Materials
- The paragraphs from "The PLM, developed in the 1980s" to "a common phenomenon in PwP" seem to provide background instead of giving information about the experimental choices. Hence, it may be moved into the Introduction.
- The paragraphs "Sit-to-stand and Stand-to-sit, "Functional reach", and "Locomotion", should be summarized to include the specific details making the reader understand the proposed framework.
- The sentence "Data were excluded in cases of marker occlusion and/or inappropriate interpolation" in Subsubsection 2.3.7 confirms that the issue of occlusions was faced when preprocessing data. Therefore, why not use inertial sensors that intrinsically prevent it? This sounds as a study limitation that should be acknowledged at the end of the Discussion; moreover, as mentioned before, the choice of optical motion tracking should be sufficiently justified in the Introduction.
- As one can deduce from Figure 1, the turning action has been performed only in one direction. Have you considered both clockwise and counterclockwise rotations? Please, clarify.
- To make the paper more readable, one table summarizing all the indexes extracted in the processing, their meaning, and the reference to their formulas, as well as possible related works from which authors drew inspiration.
- Were the 12 participants recruited in Study 2 the same as the 12 individuals recruited in Study 1? If not, it two different cohorts are included in the same study, which could have been split into two studies for the sake of clarity.
Minor comments
- If any Word template has been used, I suggest adopting the "justified" text.
- At least one reference, possibly clinical, should be reported at the end of the sentence "PwP were expected to show wider stance angles and greater stance width to improve their stability", as well as wherever the authors report an expected motor behavior of PD people. Similarly, one reference should be added when speaking of Intra-class correlation as a method for assessing potential fatigue effects.
- Figures could have been merged in a single panel by using radar plots instead of the scatter plots.
- Some sentences needed to be grammatically revised to make the text worthy of a scientific paper. For instance, "An example is the Timed Up and Go (TUG) test" could be modified as "An example is given by the Timed Up and Go (TUG) test"; "TUG's" should not report the Saxon genitive.
- Some sentences could be modified to avoid any misunderstanding: for instance, the sentence " with no motor or neurological impairments" (Subsection 2.1) could be " with no other motor or neurological impairments", since it is related to PD patients.
Round 2
Reviewer 3 Report (New Reviewer)
Comments and Suggestions for Authors
The authors have revised the manuscript by matching most of the previously highlighted points. For instance:
- grammar has been well revised in the suggested points;
- biomechanical characteristics have been well summarized in tables;
- the contents about prevalence, dopaminergic cells, and motor symptoms have been rightfully included at the beginning of the Introduction, thus giving a "comfortable" contextualizing incipit to the reader.
- music-and-movement-based intervention has been successfully included to support its usefulness in the rehabilitation context.
- the paragraphs from "The PLM, developed in the 1980s" to "a common phenomenon in PwP" have been moved into the Introduction, thus enhancing the background provided to the reader.
- the paragraphs "Sit-to-stand and Stand-to-sit, "Functional reach", and "Locomotion" have been well summarized, thus improving the paper readability.
- the issue of occlusions has been sincerely acknowledged as a study limitation at the end of the Discussion.
- the discussion about the biomechanical characteristics and the expected motor behavior of PD people has been properly supported by references.
- all the indexes extracted, their meaning, their formulas, and their related references have been thoroughly summarized in a specific Table.
- most of the Figures have been well merged in one single panel for the sake of a better readability.
- the ICC-based assessment of potential fatigue effects has been adequately introduced and supported with relevant methodological studies in Subsubsection "Biomechanical analysis"
s
However, some aspects still remain to be accomplished and/or improved to enhance the quality of the study.
The main concerns the authors should carefully address are reported point-by-point as follows.
Introduction
- Deep-brain stimulation has been mentioned, but a couple of references should be added when speaking about the invasiveness of DBS (e.g., https://doi.org/10.3390/make7030084, https://doi.org/10.1016/j.wneu.2023.07.103). Moreover, the brief insight about the MRI-based diagnosis of PD, which can be directly added in the same paragraph, still lacks in the Introduction.
- The sentence "These clinical manifestations [...] mobility, balance, and independence in daily life" could be extended by mentioning a couple of studies about PD rehabilitation, otherwise it would be perceived as out-of-context.
- The sentences "optical motion captures [...] for wider clinical application" need to be written in such a way as to strongerly justify the choice of a fixed motion capture system within the proposed framework, since the drawbacks of illumination independence, occlusion, and portability have not been adequately supported by reference yet (e.g., https://www.mdpi.com/1424-8220/25/1/260, https://doi.org/10.1007/978-3-030-01845-0_224, 10.1109/LRA.2019.2928775 ).
- At least one paragraph clearly highlighting the paucity in the literature that is filled by the presented study, and the novelty of this latter should be added.
Materials
- Even though it is clear that turning actions entail clockwise and counterclockwise directions, a brief mention to their possible impact on the results and how they have been treated in the processing should be included, since the sign of the signal from a motion tracking system would be different, thus modifying the value of the related features. In fact, this "directional issue" has already been explored in other studies with kinematic data feeding Deep Learning architectures, since they directly influence of the features extracted from the processed data.
- Since 7 of the 12 participants recruited in Study 2 are different from the ones recruited in Study 1, were these different groups matched by characteristics?
Minor comments
- Study limitations should be supported by a couple of references, if possible.
- Tables should be enhanced to be "aesthetically worth" of a scientific paper (suggested appearance: Table 1 of https://doi.org/10.3390/s25010260)
- Figure 8 merged images in a single panel, but this could be arranged horizontally to reduce the space occupation.
Author Response
Please see the attachment.

This manuscript is a resubmission of an earlier submission. The following is a list of the peer review reports and author responses from that submission.
Round 1
Reviewer 1 Report
Comments and Suggestions for Authors
The manuscript aims to use FMA-P as a comprehensive methodology to assess PwP. Authors have done an excellent job outlining every small variable that may impact the gait metrics and have done a great job going into the details of each test which was administered. Unfortunately, the manuscript lacks clinical evidence of significant difference in outcomes (improved gait metrics post interventions) using FMA-P vs motion measures individually. To prove that FMA-P is indeed a better composite score than usual individual gait variables authors require a larger sample size, rigorous clinical selection of PD patients and longitudinal follow-up to justify use of such elaborate testing methods. Although the manuscript is overall well written it tends to lack the succinctness especially in the introduction section.
A big concern is that several variables have been compared but it's unclear if multiple comparisons corrections were employed by the authors. There is a tendency to refer to "trends to significance" in sections of the paper. I am concerned of significant type I error.
Overall, aside from noting there are differences in measures along with specific tasks the paper doesn't translate to why/how FMA-P can be translated to clinical use.
Major comments:
Materials and methods-
Incorrect “n” of total healthy control (9 in Switzerland and 6 in UK vs 12 Healthy controls mentioned in first line). There is no description of inclusion/exclusion criteria of individuals within this study. Were patients excluded in case of other neurodegenerative disorders such as spinal stenosis or orthopedic problems that can affect parameters being examined.
Several parameters have been examined – was there correction for multiple corrections performed?
Section 4.1.2: “Sit to stand” – mention of “PWP took longer to transition from sitting to standing compared to controls” However the p value listed does not appear to be statistically significant. Same goes for maximum trunk inclination in AP plane (p value listed is 0.054).
Author Response
Thank you very much for taking the time to review this manuscript. Please find the detailed responses in the attachment and the corresponding revisions highlighted in the re-submitted files.

Reviewer 2 Report
Comments and Suggestions for Authors
Title:
- I believe that the title of the article could be more appropriate for the real idea of ​​the study, which is an instrument on plantar pressure. Anyone who reads the current title will not understand what this is about.
Abstract:
- I believe that the authors should include the following terms in the abstract: Introduction, objectives, methods, results and conclusions to help readers better understand the abstract text.
- In addition, it is not clear what type of study it is and how the outcome should be interpreted. Does the higher the outcome, the better the result? Or vice versa? In addition, I believe that the authors should include the values ​​of each group in the results.
- The methods do not mention which parameters will be analyzed or how they will be interpreted. However, there is mention of this in the conclusions of the abstract. I believe that the authors can improve this description in the methods of the abstract.
Introduction:
- The introduction is too long. I believe that the authors can reduce it.
- I believe that section 2, as it is called in the article, should be part of the study methods.
- Section two is very well detailed.
Methods:
- What is the study design?
- What are the study eligibility criteria?
- Table 1 cannot be in the study methods, since the methods are still describing how the volunteers were recruited and their methods. I suggest taking Table 1 to the results section.
- Why didn't the authors perform a better standardization of the sample? By pairing the participants by sex and age? This is an important bias in this study.
- How was the statistical adjustment made between the sex and age outcomes? This needs to be better detailed in the study statistics.
Author Response

(The authors gave the same response as above.)

Reviewer 3 Report
Comments and Suggestions for Authors
This paper introduces the Functional Mobility Assessment for Parkinson’s (FMA-P), a novel protocol designed for a more detailed evaluation of movement in individuals with Parkinson's disease. The study compares the FMA-P, which integrates motion capture and gait analysis during tasks mimicking daily life, with the standard Timed Up and Go (TUG) test. Findings from a pilot study with Parkinson's patients and healthy controls suggest that the FMA-P reveals subtle yet significant differences in functional mobility and motor control not always captured by traditional assessments. Specifically, the FMA-P's inclusion of a goal-directed reach task and detailed biomechanical analysis provides deeper insights into movement quality, potentially aiding in earlier diagnosis, tracking intervention effectiveness, and guiding rehabilitation strategies for Parkinson's disease.
Some General Comments:
1. The manuscript uses inclusive language, such as “people with PD.” Thanks.
2. The introduction lacks the objective of the study what is it for?. The authors must include the research question.
3. I suggest incorporating Section 2 into the methodology section, and adding figures with diagrams or schematics to better understand the procedure used.
4. In the results section, organize the information more clearly; avoid repeating in the text what is already in the tables. Perhaps include a summary table for each extracted feature.
5. In the discussion, include the limitations of the study and suggestions for future directions. Are the different methodologies comparable ?
6. The Performance Score, a qualitative assessment tool, needs to be evaluated with a larger sample size in future studies to integrate qualitative aspects fully for clinical application
Specific Comments:
• Line 239 is unclear.
• Line 250 is missing a reference.
• Subheading of Section 2.1.3 — should it be changed to “Functional Reach Test”?
• In the procedure, it is not clear whether the total circuit is 3 meters. How far is the doorway frame, and at what distance should the keys be retrieved?
• In section 4.1.2 results, ICCs are mentioned — what are their values and confidence intervals?
• In the results, why are comparisons not made between the locomotion results of the proposed tool and the TUG?
• During the execution of the TUG, was it also recorded with Vicon and a platform?
• Line 804 is missing a reference.
Author Response

(The authors gave the same response as above.)

Round 2
Reviewer 2 Report
Comments and Suggestions for Authors
This study has a serious methodological error. Evidence in the literature shows that balance and gait (functional mobility) worsen with age. This is already well established in the literature. And this study is a study that aims to evaluate mobility and compares subjects of different ages. This is a bias and a methodological problem that compromises the results of the study. For this reason, I maintain my decision to reject the study for publication. I suggest that the authors increase the sample and match the subjects in relation to sex and age group, in order to have more reliable and safe results.